

# Evaluation of the daytime tropospheric loss of 2-methylbutanal

María Asensio[1,2], María Antiñolo[1,2,†], Sergio Blázquez[2], José Albaladejo[1,2], Elena Jiménez[1,2,*]

[1]Instituto de Investigación en Combustión y Contaminación Atmosférica, Universidad de Castilla-La Mancha, Camino de Moledores s/n, Ciudad Real, 13071, Spain

[2]Departamento de Química Física, Universidad de Castilla-La Mancha, Avda. Camilo José Cela 1B, Ciudad Real, 13071, Spain

*Correspondence to*: Elena Jiménez (elena.jimenez@uclm.es)

**Abstract.** Saturated aldehydes, *e.g.* 2-methylbutanal (2MB, $CH_3CH_2CH(CH_3)C(O)H$), are emitted into the atmosphere by several biogenic sources. The first step in the daytime atmospheric degradation of 2MB involves gas-phase reactions initiated

by hydroxyl (OH) radicals, chlorine (Cl) atoms and/or sunlight. In this work, we report the rate coefficients for the gas-phase reaction of 2MB with OH ($k_{OH}$) and Cl ($k_{Cl}$) together with the photolysis rate coefficient ($J$) in the ultraviolet solar actinic region in Valencia (Spain) at different times of the day. The temperature dependence of $k_{OH}$ was described in the 263-353 K range by the following Arrhenius expression: $k_{OH}(T)=(8.88\pm0.41)\times10^{-12} \exp[(331\pm14)/T]$ cm$^3$ molecule$^{-1}$ s$^{-1}$. At 298 K, the reported $k_{OH}$ and $k_{Cl}$ are $(2.68\pm0.07)\times10^{-11}$ cm$^3$ molecule$^{-1}$ s$^{-1}$ and $(2.16\pm0.16)\times10^{-11}$ cm$^3$ molecule$^{-1}$ s$^{-1}$. Identification and

quantification of the gaseous products of the Cl-reaction and those from the photodissociation of 2MB were carried out in a smog chamber by different techniques (Fourier transform infrared spectroscopy, proton transfer time-of-flight mass spectrometry, and gas chromatography coupled to mass spectrometry). The formation and size distribution of secondary organic aerosols formed in the Cl-reaction was monitored by a fast mobility particle sizer spectrometer. A discussion on the relative importance of the first step in the daytime atmospheric degradation of 2MB is presented together with the impact of

the degradation products in marine atmospheres.

## 1 Introduction

The saturated aldehyde 2-methylbutanal (2MB, $CH_3CH_2CH(CH_3)C(O)H$) is emitted into the low atmosphere from several sources. It is known that 2MB is formed during the fermentation and drying processes of cocoa beans (Utrilla-Vázquez et al., 2020), the manufacturing process of tea leaves (*e.g.* Camellia Sinensis (Flaig et al., 2020)), and as a consequence of the stress

suffered by grapevine leaves due to drought (Griesser et al., 2015). Besides, this aldehyde is emitted into the atmosphere by wildland fires (Urbanski et al., 2008) and many industrial activities, such as poultry rendering operations when animal by-products are processed (Kolar and Kastner, 2010). As a secondary pollutant, 2MB can be formed in situ in the atmosphere by

---

† Currently at *Escuela* de *Ingeniería Industrial* y *Aeroespacial*. Universidad de Castilla-La Mancha. Avenida Carlos III s/n. Real Fábrica de Armas. 45071 Toledo (Spain).



oxidation of 2-methyl-1-butanol ($CH_3CH_2CH(CH_3)CH_2OH$), which is used as a biochemical pesticide, commercially used as a solvent in paints and oils and as flavorant in many processed foods.

Once in the atmosphere, 2MB can react with diurnal tropospheric oxidants, such as hydroxyl (OH) radicals, chlorine atoms (Cl) in coastal or marine regions or ozone ($O_3$) in polluted environments. During daytime, 2MB can also be photolyzed by the solar actinic radiation ($\lambda > 290$ nm). Since the gas-phase chemistry of 2MB can lead to the formation of secondary pollutants, its emission may have a significant impact on tropospheric chemistry and air pollution at a local/regional scale. For that reason, in this work we evaluate the potential impact of the diurnal degradation of 2MB. Firstly, the photodissociation of 2MB

(Reaction R1) has been investigated at room temperature by determining its absorption cross sections ($\sigma_\lambda$) between 220 and 360 nm, the photolysis rate coefficient ($J$) under the irradiation conditions of this work ($\lambda \geq 290$ nm) and the corresponding effective quantum yield ($\phi_{eff}$).

$CH_3CH_2CH(CH_3)C(O)H + h\nu$ ($\lambda \geq 290$ nm) $\rightarrow$ Products          $J$                                    (R1)

Secondly, the gas-phase kinetics with OH (Reaction R2) and Cl (Reaction R3) under tropospheric conditions of temperature

and pressure was investigated to assess the tropospheric lifetime ($\tau$) of 2MB due to both removal routes.

$OH + CH_3CH_2CH(CH_3)C(O)H \rightarrow$ Products          $k_{OH}$                                 (R2)

$Cl + CH_3CH_2CH(CH_3)C(O)H \rightarrow$ Products          $k_{Cl}$                                  (R3)

The rate coefficient $k_{OH}$ was determined between 263 and 353 K as a function of total pressure (50-600 Torr of He), while $k_{Cl}$ was measured at 298 K and (760±5) Torr of air. Finally, the gaseous products of reactions R1 and R3 have been identified

under $NO_x$-free conditions using several detection techniques and a reaction mechanism is proposed for clean atmospheres. Furthermore, the formation yield of secondary organic aerosols (SOAs) formed in reaction R3 has been measured to assess the impact of 2MB on the formation of ultrafine particles. With all this information, we discuss the potential impact of atmospheric 2MB on local or regional air quality in terms of the estimated tropospheric lifetime and the reaction products formed.

## 2 Experimental methods

In this section, a brief description of the experimental techniques and the methodology employed in this work is given. More details can be found in the Supporting Information (SI).

### 2.1 Photodissociation of 2-methylbutanal

### 2.1.1 Gas-phase ultraviolet (UV) absorption spectroscopy (220-360 nm)

Ultraviolet absorption spectroscopy was used to determine the absorption cross sections of 2MB as a function of wavelength

($\sigma_\lambda$ in base $e$) between 220 and 360 nm. The experimental setup employed in this work has been described in detailed elsewhere (Blázquez et al., 2020). This system consists of a deuterium-tungsten light source (DT-200, StellarNet) placed at the entrance



of a 107.15 cm jacketed Pyrex® cell, connected by an optical fibre to a f/2 spectrometer which poses a concave holographic grating (590 grooves/mm) and a 2048 pixel CCD camera (BLACK-Comet model C, StellarNet). The absorbance (in base 10) is recorded in a computer with the data acquisition software (SpectraWiz v5.33). The experiments were carried out by

introducing pure gaseous 2MB (1.085 – 6.642 Torr) into the UV cell in static mode. Applying the Beer-Lambert's law, $\sigma_\lambda$ was determined from the slope of the absorbance (in base $e$) versus 2MB concentration ([2MB]). **Figure S1** of the SI shows some examples of the absorbance *versus* [2MB]. Despite the employed spectrometer was able to acquire data with a higher resolution, $\sigma_\lambda$ values are given at every nanometre for an easier presentation (see **Table S1**).

**2.1.2 Continuous irradiation with a solar simulator (λ≥290 nm)**

A schematic of the set-up used in this work to investigate the photochemistry of 2MB under atmospheric conditions is shown in **Fig. 1**. A Pyrex ($l$=20 cm and $\phi$=4 cm) cell sealed with quartz windows was filled with (760 ± 3) Torr of diluted 2MB (interval of dilution factor from $6.96\times10^{-4}$ to $1.92\times10^{-3}$ in synthetic air) from a 10-L Pyrex storage bulb at (298 ± 2) K. The partial pressure of 2MB, the total pressure inside the storage bulb and the pressure in the photolysis cell were measured by

capacitance pressure transducers (Leybold, model Ceravac, 10 and 1000 Torr full scale). The initial concentration of 2MB in the photolysis cell ranged from 1.2 to $6.9\times10^{16}$ molecule cm$^{-3}$, measured by FTIR spectroscopy. Before irradiating the sample and after each irradiation time, the Fourier transform infrared (FTIR) spectrum of the mixture was recorded in a 16-L White-type cell with an optical path length of 96 m by a FTIR spectrometer (Thermo Fischer Scientific, model Nicolet Nexus 870, Madison, WI, USA) with a liquid $N_2$-cooled MCT (Mercury Cadmium Telluride) detector. IR spectra were recorded between

650 and 4000 cm$^{-1}$ at a resolution of 2 cm$^{-1}$, after the accumulation of 32 interferograms. The selected IR band for monitoring 2MB was the one between 2600 and 3000 cm$^{-1}$.

The sample of 2MB in air was irradiated by an ABA class solar simulator (SunLite$^{TM}$ Solar Simulator, model 11002-2) during 30, 60, 90, 120, and 150 min. The solar simulator, equipped with a Xe arc lamp and an Air Mass (AM) 1.5G filter, emits radiation over 290 nm with a spectrum that simulates the solar reference spectrum AM 1.5G. This reference spectrum

corresponds to the terrestrial solar spectral irradiance on a surface when the air mass is 1.5 and the sun is about 41° above the horizon and under specific atmospheric conditions defined by the U.S. Standard Atmosphere. Irradiance was determined in each experiment with a 1-cm$^2$ solar reference cell for which a 100 mV output corresponds to 1 Sun, defined as the irradiance of the AM1.5G reference solar spectrum described above. In this work, the measured irradiance was (2.225±0.145) Suns.

As shown in **Fig. 1**, during the irradiation of the diluted sample of 2MB, control valves 2 and 3 were closed. After irradiation, the gas sample was expanded to the 16-L FTIR cell to measure the 2MB concentration over time ([2MB]$_t$) and, thus, to determine the photolysis rate coefficient, $J$. In that case, the total pressure decreases to 7 Torr and control valves 1 and 4 were closed.



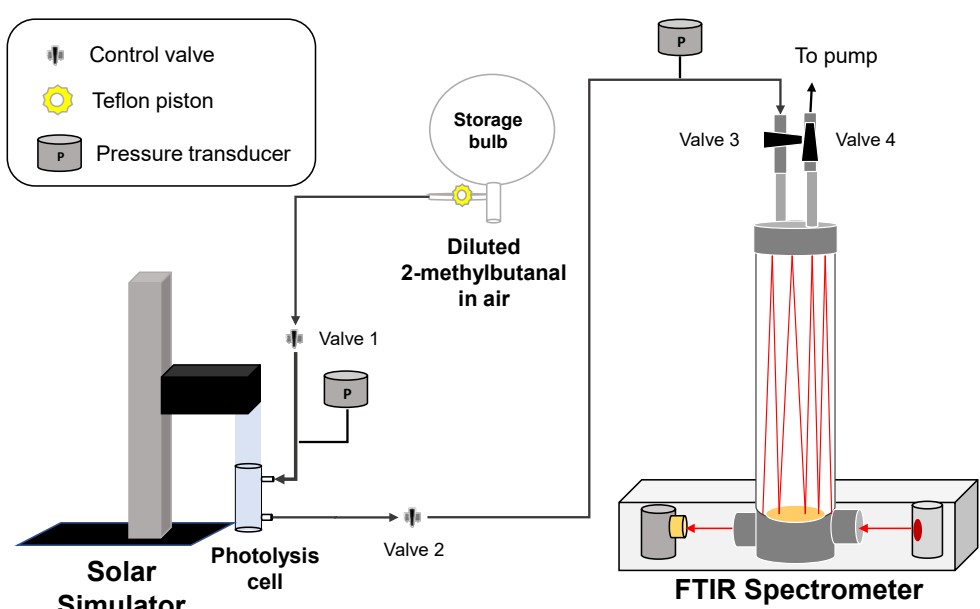

**Figure 1.** Schematics of the photolysis set-up using a solar simulator as a continuous irradiation source:

Possible losses due to the heterogeneous reaction of 2MB onto the walls were evaluated in experiments without light at different times (45, 90 and 135 min). As the photolysis cell used has a surface/volume ratio (276.46 cm$^2$/251.33 cm$^3$= 1.1 cm$^{-1}$), the heterogeneous loss of 2MB contributes significantly to the total decay of 2MB. The heterogeneous loss rate, $k_{heterog}$ (in s$^{-1}$), has been measured to contribute around 24% to the total loss. Therefore, the photolysis rate coefficient of 2-methylbutanal at $\lambda > 290$ nm, $J$ (in s$^{-1}$), can be obtained from the slope of the plot of $ln([2MB]_0/[2MB]_t)$ versus time according to Eq. (1).

$$ln([2MB]_0/[2MB]_t) = (k_{heterog}+J)t \qquad (1)$$

For the identification of the photolysis products, three complementary detection techniques were used: FTIR spectroscopy, gas chromatography-mass spectrometry (GC-MS) and proton transfer reaction time-of-flight mass spectrometry (PTR-ToF-MS). The gas chromatography-mass spectrometer (Thermo Electron, models Trace GC Ultra and DSQ II) was equipped with a BPX35 column (30 m × 0.25 mm ID × 0.25 μm, SGE Analytical Science) working at a temperature ramp that ranged between 40 and 250 ℃ (Ballesteros et al., 2017). The solid-phase microextraction technique was used as a sampling method in these experiments, thus, a 50/30 μm divinylbenzene/carboxen/polydimethylsiloxane (DVB/CAR/PDMS) fibre (Supelco) was exposed during 10 min to the gas mixture of the photolysis cell and then, the fibre was injected into the GC-MS during 5 min. Independently, in some experiments a PTR-ToF-MS (IONICON, model PTR-TOF 4000), working with a field density ratio ($E/N$) of 137 Td, a time resolution of 20 s, and a detected mass range between 29 and 390.86 a.m.u. (Antiñolo et al., 2010) was coupled to the FTIR cell, in which the content of the photolysis cell was diluted in 760 Torr of air, to detect the photolysis products at the end of the experiment. However, quantification of the photolysis products could only be carried out by FTIR



spectroscopy, as sampling with the PTR-ToF-MS from the volume of the FTIR cell made the pressure drop rapidly and it did not allow to have a stable signal. Furthermore, GC-MS quantification was not performed since no internal standards were added to the 2MB mixture.

### 2.2 Pulsed laser photolysis coupled to laser induced fluorescence (PLP-LIF) technique


The absolute kinetic study of the reaction of OH radicals with 2MB (reaction R2) was performed as a function of temperature (T = 263-353 K) and total pressure (P = 50-600 Torr of He) in a Pyrex doubled-wall reactor of *ca.* 200 mL (Martínez et al., 1999). Briefly, OH radicals were generated *in situ* from the pulsed laser photolysis (PLP) by a KrF excimer laser (Coherent, ExciStar 200) at 248 nm (laser fluence of 23 mJ pulse$^{-1}$ cm$^{-2}$ at 10 Hz) of a suitable precursor ($H_2O_2$ or $HNO_3$). The OH radicals generated in the electronic ground state were excited at *ca.* 282 nm (0.4–1.0 mJ pulse$^{-1}$ at 10 Hz) to the first excited electronic state by a tuneable pulsed laser (Continuum, ND60; pumped by Continuum, NY 81 CA-10). The photolysis and excitation lasers traverse the reactor perpendicularly. From the intersection of both lasers, in which reaction takes place, the excited OH radicals emit off-resonance laser induced fluorescence (LIF) (at *ca.* 310 nm), which was collected by a photomultiplier tube (Thorn EMI, 9813B model) set perpendicular to both lasers. The gas temperature (T) was measured by a type K (chromel–alumel) thermocouple inserted several millimetres above the reaction and T was kept constant (±0.2 K) during each experiment. Under *pseudo*-first order conditions (*i.e.,* when the initial concentration of OH-precursor and 2MB are in large excess with respect to that of OH radicals), the time evolution of the LIF signal follows a single exponential expression (see some examples in **Fig. S2**). From the analysis of these decays, the *pseudo*-first order rate coefficient, $k'$, was determined at given [OH-precursor]$_0$ and [2MB]$_0$ for each temperature and pressure. In the absence of 2MB, the measured *pseudo*-first order rate coefficient is $k_0'$. Examples of the $k'$-$k'_0$ *versus* [2MB]$_0$ plots at 263 and 353 K, from which the rate coefficient $k_{OH}$ was obtained, are depicted in **Fig. S3**.




The experimental conditions used in the absolute kinetics experiments on the OH+2MB reaction are listed in **Table S2** of the SI. The tabulated parameters are the total mass flow rate ($F_{total}$), which includes the sum of the calibrated flow rates of 2MB (between 3.1 and 29 sccm), He and OH-precursor (between 14.4 and 96.8 sccm), the total cell pressure ($P_{cell}$), and temperature. The ranges of [OH-precursor]$_0$, [2MB]$_0$, dilution factor $f$ of the mixture bulb, and $k'$ are also listed for each temperature and pressure. [OH-precursor]$_0$ was measured by UV absorption spectroscopy between 200 and 230 nm using the same experimental systems described in Sect. 2.1.1. In this case, the absorption measurements were carried out under flowing conditions. From [OH-precursor]$_0$ and the fluence at 248 nm, the initial amount of OH radicals inside the reactor cell was estimated. More details in the SI.


### 2.3. Simulation (*smog*) chambers coupled to different detectors


For the kinetic and mechanistic study of the Cl-reaction with 2MB, several experimental systems were used (Ballesteros et al., 2017; Antiñolo et al., 2019; Antiñolo et al., 2020). Two cylindrical chambers made of Pyrex were available to perform the



different experiments: a 16-L cell and 264-L reactor. Both chambers were surrounded by actinic lamps (Philips Actinic BL TL 40W/10 1SL/25, $\lambda$= 340–400 nm): four for the 16-L gas cell and eight for the 264-L. These lamps were used to continuously

generate Cl atoms *in situ* by UV photolysis of Cl$_2$. The gas-phase species (2MB and Cl$_2$) were introduced in the reactor from a gas-line, and they were diluted with synthetic air at (298±2) K and (760±5) Torr of total pressure. The total pressure in the gas-line and in the reactors was measured with two capacitor pressure transducers (Leybold, model Ceravac, 10- and 1000-Torr full scale).

Three different detection methods were employed: FTIR spectroscopy, GC-MS and PTR-ToF-MS, previously described in
Sect. 2.1.2.

### 2.3.1. Relative kinetic studies (298 K and 760 Torr of air)

The kinetic experiments were carried out by mixing 2MB, a reference compound (ethanol or isoprene) and Cl$_2$ in the 16-L cell and using FTIR to monitor 2MB and the reference compound as a function of reaction time. The IR bands selected for
monitoring the loss of 2MB and the reference compounds were centred at 2700 cm$^{-1}$ for 2MB, 1070 cm$^{-1}$ for ethanol and 3095 cm$^{-1}$ for isoprene. The disappearance of 2MB and the reference compound is mainly due to reaction with Cl, although they can also be lost by heterogeneous reaction onto the reactor walls, UV photolysis and/or reaction with the oxidant precursor. These loss processes were evaluated in preliminary and independent experiments in the absence of Cl$_2$ and UV light (wall loss, $k_w$), in the absence of UV light (reaction with Cl$_2$, $k'_{Cl_2} = k_{Cl_2}[Cl_2]_0$) and irradiating in the absence of Cl$_2$ (UV photolysis, $k_{h\nu}$) as
described in previous studies (Antiñolo et al., 2019; Antiñolo et al., 2020). **Table 1** summarizes the loss rate coefficients for these processes and the overall loss rate coefficients, $k_{Total\_loss}$ ($k_{loss}$ for 2MB and $k_{Ref,loss}$ for the reference compound).

**Table 1.** Measured loss rate coefficients of 2MB and the reference compounds.

| Compound | $k_w$ / $10^{-4}$ s$^{-1}$ | $k'_{Cl_2}$/ $10^{-5}$ s$^{-1}$ | $k_{h\nu}$/ $10^{-5}$ s$^{-1}$ | $k_{Total\_loss}$/ $10^{-4}$ s$^{-1}$ |
|---|---|---|---|---|
| 2MB | $3.20 \pm 1.15$ | negl. | negl. | $3.20 \pm 1.15$ |
| Ethanol | $10.7 \pm 0.14$ | negl. | negl. | $10.7 \pm 0.14$ |
| Isoprene | $2.27 \pm 0.99$ | $9.54 \pm 0.54$ | negl. | $11.8 \pm 1.13$ |

negl. negligible


Therefore, taking into account the overall losses for both 2MB and the reference compound, the integrated rate equation is given by the following expression:

$$\ln\left(\frac{[2MB]_0}{[2MB]_t}\right) - k_{loss}t = \frac{k_{Cl}}{k_{ref}}\left[\ln\left(\frac{[Ref]_0}{[Ref]_t}\right) - k_{Ref,loss}t\right] \qquad (2)$$




where $k_{ref}$ is the rate coefficient for the Cl-reaction with the reference compound at 298 K and 760 Torr. $[2MB]_0$, $[2MB]_t$, $[Ref]_0$ and $[Ref]_t$ are the concentrations of 2MB and the reference compound at the beginning of the reaction and at a reaction time $t$, respectively. Initial concentrations in the cell were $[2MB]_0 = (4.9 – 6.6) \times 10^{14}$ molecule cm$^{-3}$, $[Cl_2]_0 = (3.3 – 5.9) \times 10^{14}$ molecule cm$^{-3}$, $[ethanol]_0 = (4.6 – 3.6) \times 10^{14}$ molecule cm$^{-3}$ and $[isoprene]_0 = (5.1 – 5.7) \times 10^{14}$ molecule cm$^{-3}$. In **Fig. S4**, an example of the plots of Eq. (2) is presented for both reference compounds.

### 2.3.2. Product study in the Cl reaction

***Gaseous products.*** The identification of the gaseous products generated in the reaction of 2-methylbutanal with Cl atoms was performed using complementary detection techniques: FTIR spectroscopy (using the 16-L reactor) (Ballesteros et al., 2017; Antiñolo et al., 2019), GC-MS, and PTR-ToF-MS (using the 264-L simulation chamber) (Antiñolo et al., 2020) in separate experiments. In all cases, preliminary tests were carried out to check if products were generated during the dark reaction of 2MB with $Cl_2$ and/or during the UV light exposure of 2MB. When GC-MS was used, no products due to these processes were observed. Nevertheless, with FTIR and PTR-ToF-MS the formation of some products was observed during UV light exposure of 2MB (see Sect. 4.3.1). In all the experiments, the reaction mixture was irradiated during 60 min. IR spectra, chromatograms, and mass spectra were recorded every 2 min, 10 min and 20 sec respectively. In **Table 2**, the initial concentrations in the 2MB/ $Cl_2$/air mixtures are summarized. Because of the high sensitivity of the PTR-ToF-MS, the initial concentrations of 2MB and $Cl_2$ were reduced with respect to those employed in the FTIR and GC-MS experiments. Furthermore, at the inlet of the PTR-ToF-MS the reaction mixture was diluted (a factor of 1/5) with an air flow by means a dynamic inlet dilution system to avoid signal saturation.

**Table 2.** Initial concentrations of 2MB and Cl-precursor in the reactor.

| Detection method | $[2MB]_0/10^{14}$ molecules cm$^{-3}$ | $[Cl_2]_0/10^{14}$ molecules cm$^{-3}$ |
| --- | --- | --- |
| FTIR | 3.6–6.2 | 2.2-12 |
| GC-MS | 6.2-8.8 | 6.5-8.5 |
| PTR-ToF-MS | 0.18-0.29 | 0.24-0.27 |

***Secondary organic aerosols (SOAs):*** For the detection and quantification of the formation of SOAs in the Cl-reaction of 2MB, both the 16-L cell and the 264-L reactor were connected and simultaneously used as described previously (Antiñolo et al., 2020). The concentration of gaseous 2MB was monitored every 2 min by the FTIR spectrometer described in Sect. 2.1.2, whereas the formed SOAs were monitored by a Fast Mobility Particle Sizer (FMPS) spectrometer (TSI 3091). The FMPS spectrometer measures the particle size distribution in the system between 5.6 and 560 nm and every 1 s (although data were averaged for 1 min in this work). A 2MB/$Cl_2$/air mixture was introduced at $(760 \pm 5)$ Torr and $(299 \pm 2)$ K in the reactors with concentrations of 2MB and $Cl_2$ in the $(5.5 – 13) \times 10^{14}$ molecule cm$^{-3}$ and $(4.8 – 10) \times 10^{14}$ molecule cm$^{-3}$ ranges, respectively. The SOA formation and the loss of 2MB loss was continuously monitored during about 1 hour. During the first 15 min, the gas mixture was kept in the dark to check for SOA formation by dark reactions; then, the lights surrounding the 264-L reactor




were turned on for 30-40 min to measure the SOA formed in reaction (R3); finally, the lights were switched off to evaluate the

loss of the SOA formed due to the walls or other dark processes during 15 min. The SOA formation yield, $Y_{SOA}$, can be defined

as the ratio between the aerosol mass formed, $M_{SOA}$, considering a 1.4 µg m$^{-3}$ density for the SOA (Hallquist et al., 2009), and

the 2MB lost in reaction (R3), $\Delta[2MB]$, (Equation (3)).

$$Y_{SOA} = \frac{M_{SOA}}{\Delta[2MB]} , \qquad (3)$$

Both parameters, $M_{SOA}$ and $\Delta[2MB]$ were corrected by accounting for their loss as described previously (Antiñolo et al., 2019),

and $Y_{SOA}$ was determined from $M_{SOA}$ *versus* $\Delta[2MB]$ under different $[2MB]_0$ (see some examples in **Fig. S5**).

**3. Results and discussion**

**3.1 UV Photochemistry of 2-methylbutanal**

**3.1.1 Photolysis frequency (*J*) and effective quantum yield ($\Phi_{eff}$) at λ≥290 nm**

As shown in **Fig. 2**, 2MB absorbs in the ultraviolet range, exhibiting a weak absorption band in the 220-360 nm range, due to

the forbidden $n$-$\pi^*$ transition of the C=O chromophore, with a maximum at 296 nm. The maximum $\sigma_\lambda$ was determined to be

$(6.25 \pm 0.08) \times 10^{-20}$ cm$^2$ molecule$^{-1}$ (stated uncertainty in our results throughout all the document is ±2σ statistical). Therefore,

in the troposphere the actinic radiation (λ ≥ 290 nm) can initiate photochemical processes for 2MB. Four photolysis

experiments were performed under the conditions of the present as described in Sect. 2.1.2. In **Fig. 3** the average values of

individual $ln([2MB]_0/[2MB]_t)$ obtained in the 4 experiments is plotted against $t$, showing a good linearity. From the slope of

such a plot and correcting it with the wall losses of 2MB, $J = (1.96 \pm 0.32) \times 10^{-5}$ s$^{-1}$ was obtained.

The photolysis quantum yield at a single wavelength ($\Phi_\lambda$) is related to the photolysis rate coefficient as follows:

$$J = \int_{\lambda_1}^{\lambda_2} \Phi_\lambda \ \sigma_\lambda I_\lambda d\lambda \qquad (4)$$

where $I_\lambda$ is the irradiance in photons cm$^{-2}$ nm$^{-1}$ s$^{-1}$ at λ. However, $J$ can be approximated to:

$$J \cong \Phi_{eff} \sum_{\lambda_1}^{\lambda_2} I_\lambda \sigma_\lambda \Delta\lambda \qquad (5)$$

where, in this work, $\Phi_{eff}$ is the effective quantum yield of 2MB, $I_{\lambda>290 \ nm}$ is the measured irradiance at each wavelength, $\sigma_\lambda$ is

the experimentally determined UV absorption cross sections of 2MB, and $\Delta\lambda = 1$ nm. Taking into account all these parameters,

$\Phi_{eff} = (0.30 \pm 0.05)$. This value is *ca.* half of the previously reported (Wenger, 2006) measured in the atmospheric simulation

chamber EUPHORE (Valencia) under natural irradiation conditions, $\Phi_{eff} = (0.72 \pm 0.03)$. The reason for this difference is

unknown, although, in the same study, for structurally similar aldehydes, like pentanal or 3-methybutanal, the reported $\Phi_{eff}$

were closer ((0.30 ± 0.02), and (0.27 ± 0.01), respectively) to the determined in this work.

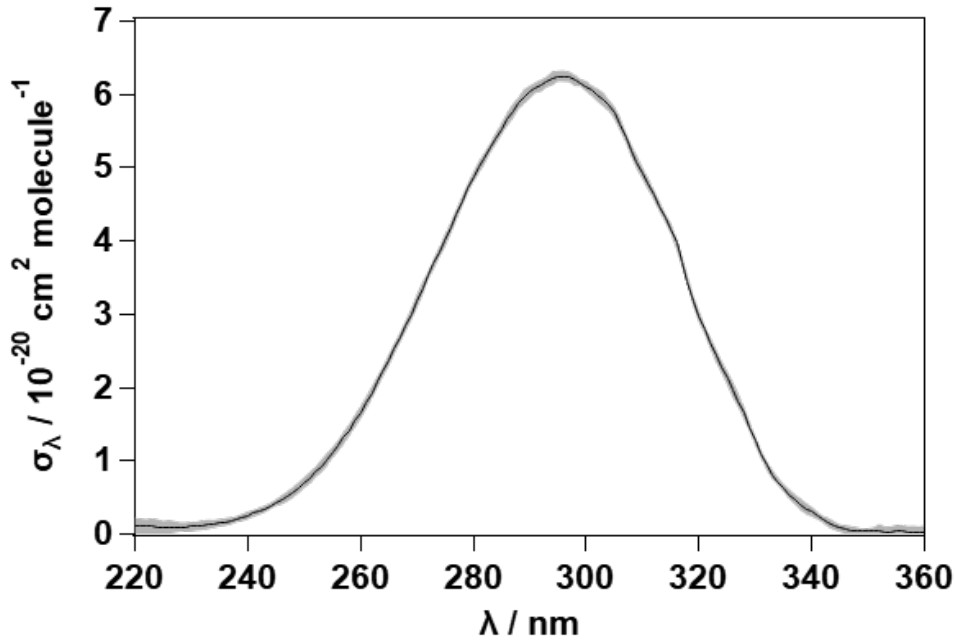

**Figure 2**. UV absorption cross sections of 2MB at 298 K. The grey shadow represents the statistical uncertainty.


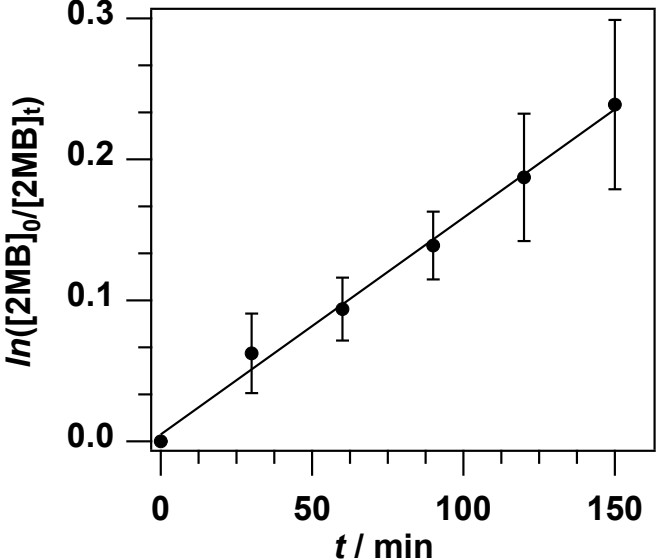

**Figure 3.** Plot of Eq. (1) in which the average $ln([2MB]_0/[2MB]_t)$ over time has been represented.




### 3.1.2. UV Photolysis products of 2MB

Photodissociation of 2MB by UV radiation can proceed through the following channels:

$$CH_3CH_2CH(CH_3)C(O)H + h\upsilon(\lambda=220\text{-}360 \text{ nm}) \rightarrow CH_3CH_2CH(CH_3)C(O) + H \qquad (R1a)$$

$$\rightarrow CH_3CH_2CH(CH_3) + HC(O) \qquad (R1b)$$

$$\rightarrow CH_3CH_2CH_2CH_3 + CO \qquad (R1c)$$

Channels (R1a) and (R1b) are radical-forming channels, while channel (R1c) is the molecular one yielding butane ($CH_3CH_2CH_2CH_3$) and carbon monoxide (CO). **Figure 4a** shows the recorded IR spectra of the 2MB/air sample before irradiation. After 150 min of irradiation, the IR features from 2MB were subtracted to identify the formed products, as shown in **Fig. 4b**. Products from channel (R1c), butane and CO, were clearly identified at 2800 - 3000 cm$^{-1}$ and 2000 - 2300 cm$^{-1}$ respectively, indicating that this channel is open in the investigated wavelength range. The product yield ($Y_{product}$) is obtained as the slope of [Product] versus the consumed 2MB, $\Delta$[2MB]. To determine the effective quantum yield of channel (R1c), $\phi_{1c}$, the yield of formation of CO ($Y_{CO}$) or butane ($Y_{butane}$) could be used if no secondary chemistry were occurring, which is not the case for CO, since it is a very end degradation product and can be formed in many oxidation reactions. Moreover, $Y_{CO}$ showed a clear dependence on the initial concentration of 2MB, increasing when the initial concentration of 2MB was low. Butane was quantified using the reference spectra shown in **Fig. S6** and the (2800-3000) cm$^{-1}$ IR band, yielding $Y_{butane} = (0.098\pm0.003)$.

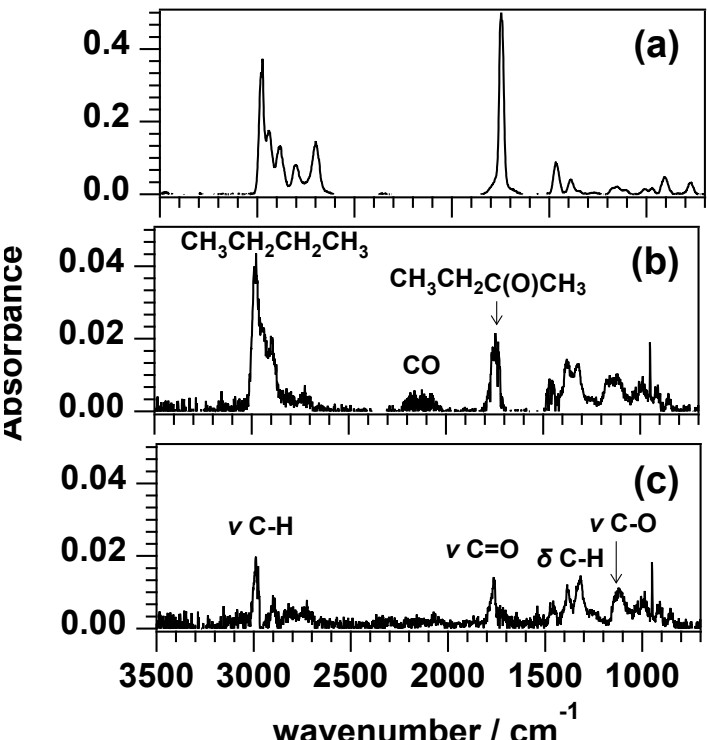

**Figure 4.** FTIR spectra of a 2-methylbutanal/air mixture (a) before
irradiation; (b) after 150 min of photolysis with the features of 2MB
subtracted (major products are labeled in the spectrum, exact
positions of the bands are given in the main text) and (c) residual
spectrum after the subtraction of the reference spectra of the
identified products shown in Fig. S6.

As presented in **Fig. 4b**, butanone ($CH_3CH_2C(O)CH_3$) was also identified as a product of 2MB photolysis in the presence of
air and its product yield was $Y_{butanone} = (0.15 \pm 0.01)$. Quantification of butanone was done using the reference spectra shown in
**Fig. S6** for the IR bands: (2850-3100) $cm^{-1}$ and (1650-1800) $cm^{-1}$. After subtracting the IR features of CO, butane and butanone,
some IR features were left in the residual spectrum (**Figure 4c**). The band centered around 1040 $cm^{-1}$ can be assigned to the
C-O stretching mode of an alcohol (probably 2-butanol or methanol according to the proposed mechanism in Sect. 3.4.2) and
the one centered around 1740 $cm^{-1}$ can be assigned to the C=O stretching mode of a carbonyl compound, such an aldehyde or
ketone. The bands centered around 3000-2860 $cm^{-1}$ can be assigned to the $Csp^3$-H stretching mode and finally, the bands
located around 1300-1400 $cm^{-1}$ can correspond to the C-H bending mode. Therefore, the remaining IR bands in the residual
spectrum are probably the result of the absorption of a mixture of oxidation products.

Butanone was also identified by GC-MS and PTR-Tof-MS. **Figure S7** shows an example of the chromatogram of a mixture
of 2MB in synthetic air, before and after photolysis (150 min). Although the PTR-ToF-MS technique allows the measurement



of VOCs in gaseous samples as a function of time by taking some flow from the sample, the volume of the photolysis cell was too small to maintain the pressure during the photolysis experiments. For that reason, a measurement after 150 min of photolysis was done during 5 min in which sample was taken from a diluted photolyzed mixture prepared in the 16-L cell only

to get the mass spectrum. Under these conditions, butanone ($C_4H_8OH^+$, m/z = 73.06) was detected.

### 3.2 Kinetics of 2MB with OH at T and P-conditions of the troposphere

The individual rate coefficient for the 2MB+OH reaction obtained at a certain temperature and total cell pressure ($P_{cell}$) are listed in **Table 3**. No pressure dependence of $k_{OH}(T)$ in the studied temperature range was observed, within the experimental

uncertainties. For that reason, all $k'$-$k'_0$ values obtained at different total pressures were combined and plotted versus $[2MB]_0$, according to Eq. (ES3) of the SI, as shown in **Fig. S3** for 263 K and 353 K. The resulting $k_{OH}(T)$ are listed in the last column of **Table 3**. We observe a slight negative T-dependence of $k_{OH}(T)$, increasing 40 % from 353 K to 263 K. In **Fig. 5**, $k_{OH}(T)$ as a function of temperature is depicted, together with the previous reported data by D'Anna et al. (2001) at room temperature. The observed T-dependence is well-described by the following Arrhenius equation (solid line in **Fig. 5**):

$$k_{OH}(T=263\text{-}353 \text{ K}) = (8.88\pm0.81)\times10^{-12} \exp[(331\pm27)/ T] \text{ cm}^3 \text{ molecule}^{-1} \text{ s}^{-1} \qquad (6)$$

where the activation energy is –(2.75 ± 0.23) kJ mol$^{-1}$. The reported $k_{OH}$ by D'Anna et al. (2001) at room temperature was $(3.28\pm0.09)\times10^{-11}$ cm$^3$ molecule$^{-1}$ s$^{-1}$, which is 22 % higher than the obtained in this work, $k_{OH}$(298 K) = $(2.68\pm0.07)\times10^{-11}$ cm$^3$ molecule$^{-1}$ s$^{-1}$. The reason for this discrepancy may be found in the different methods used: D'Anna et al. (2001) used a relative kinetic method with FTIR spectroscopy as detection technique in which they used air as bath gas at (298 ± 2) K and (760 ± 7)

Torr. They used only one reference compound for their analysis, 1-butene, and that may result in an important source of error. In addition, the selected IR band for 1-butene (3140-3070 cm$^{-1}$) is too weak to be accurately monitored. Another possibility is that this band, or the selected one for 2MB (2740 – 2670 cm$^{-1}$), could have suffered any interference by reaction products.




**Table 3.** Individual rate coefficients of the 2MB + OH reaction as a function of temperature and pressure.

| T / K | $P_{cell}$ / Torr | $k_{OH}(T)/ 10^{-11}$ cm³ molecule⁻¹ s⁻¹ | $k_{OH}(T)*/ 10^{-11}$ cm³ molecule⁻¹ s⁻¹ |
|---|---|---|---|
| 263 | 50 | 3.15 ± 0.13 | 3.18 ± 0.10 |
|  | 300 | 3.08 ± 0.15 |  |
|  | 600 | 3.31 ± 0.13 |  |
| 268 | 50 | 3.03 ± 0.22 | 3.03 ± 0.17 |
|  | 300 | 3.12 ± 0.15 |  |
| 278 | 50 | 2.74 ± 0.16 | 2.87 ± 0.15 |
|  | 300 | 3.02 ± 0.20 |  |
|  | 300 | 2.89 ± 0.24 |  |
| 288 | 50 | 2.65 ± 0.22 | 2.78 ± 0.17 |
|  | 300 | 2.84 ± 0.28 |  |
|  | 300 | 2.98 ± 0.17 |  |
| 298 | 50 | 2.83 ± 0.12 | 2.68 ± 0.07 |
|  | 50 | 2.76 ± 0.10 |  |
|  | 300 | 2.66 ± 0.11 |  |
|  | 600 | 2.62 ± 0.11 |  |
| 309 | 50 | 2.68 ± 0.18 | 2.64 ± 0.13 |
|  | 300 | 2.60 ± 0.15 |  |
| 323 | 50 | 2.46 ± 0.10 | 2.46 ± 0.08 |
|  | 300 | 2.46 ± 0.13 |  |
| 338 | 50 | 2.38 ± 0.07 | 2.35 ± 0.06 |
|  | 300 | 2.31 ± 0.10 |  |
| 353 | 50 | 2.25 ± 0.09 | 2.27 ± 0.09 |
|  | 300 | 2.40 ± 0.10 |  |
|  | 600 | 2.22 ± 0.07 |  |

* Values obtained from the $k'-k_0'$ versus $[2MB]_0$ plot combining all kinetic data at different total pressures



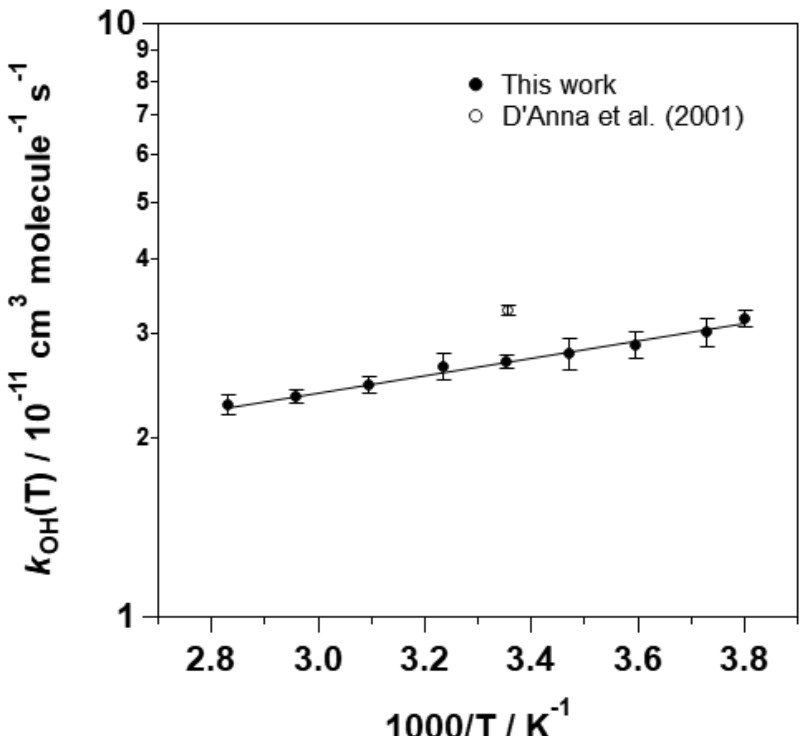

**Figure 5**. Arrhenius plot for the rate coefficient of the 2MB + OH reaction between 263 and 353 K.

### 3.3 Cl-reaction of 2MB at T and P-conditions of the marine boundary layer

### 3.3.1 Kinetics with Cl atoms

The plots of Eq. (2) for the two references used in this work can be seen in **Fig. S4**, in which a good linearity is shown, indicating that no secondary reactions were interfering. From the slope of these plots, $k/k_{Ref}$ was obtained according to Eq. (2). Thus, from the rate coefficients of the reference compounds previously reported for the Cl reaction of ethanol and isoprene (Atkinson et al., 2006; Orlando et al., 2003), the rate coefficient of 2MB with Cl ($k_{Cl}$) was determined. **Table 4** shows the rate coefficients obtained in this work with each reference compound and the averaged value, $(2.16 \pm 0.16) \times 10^{-10}$ cm$^3$ molecule$^{-1}$ s$^{-1}$. The uncertainty in $k_{Cl}$ includes the propagation of the reported errors in $k_{Ref}$, the uncertainties in $k_{loss}$ and the statistical errors from the slope of the plots shown in **Fig. S4**.




**Table 4.** Results obtained in the kinetic experiments for the gas-phase reaction of Cl with 2-methylbutanal at
298 ± 2 K and 760 ± 5 Torr of air.

| Reference | $k_{Cl}/k_{Ref}$ | $k_{Ref}$ / $10^{-10}$ cm$^3$ molecule$^{-1}$ s$^{-1}$ | $k_{Cl}$ / $10^{-10}$ cm$^3$ molecule$^{-1}$ s$^{-1}$ |
|---|---|---|---|
| Ethanol | 2.140 ± 0.038 | 1.00 ± 0.06 [1] | 2.14 ± 0.13 |
| Isoprene | 0.509 ± 0.003 | 4.30 ± 0.58 [2] | 2.19 ± 0.29 |
| **Average** | | | **2.16 ± 0.16** |

[1] Atkinson et al. (2006), [2] Orlando et al. (2003)

### 3.3.2 Identification and quantification of the gaseous products of the Cl + 2MB reaction

*Identification by GC-MS:* **Figure 6** shows the obtained chromatograms before and after 60 min reaction time and **Fig. S8** shows the mass spectra of the detected products corresponding to those chromatographic peaks. The peak corresponding to 2MB was observed at a retention time (RT) of 3.76 min. The rest of the peaks that appear in the chromatogram were assigned, according to their mass spectrum, to the following products: acetaldehyde (RT = 2.15 min), 2-butanol (RT = 3.08 min), butanone (RT = 3.21 min), methylglyoxal (RT = 4.44 min), and 2-methylbutanoic acid (RT = 5.55 min).

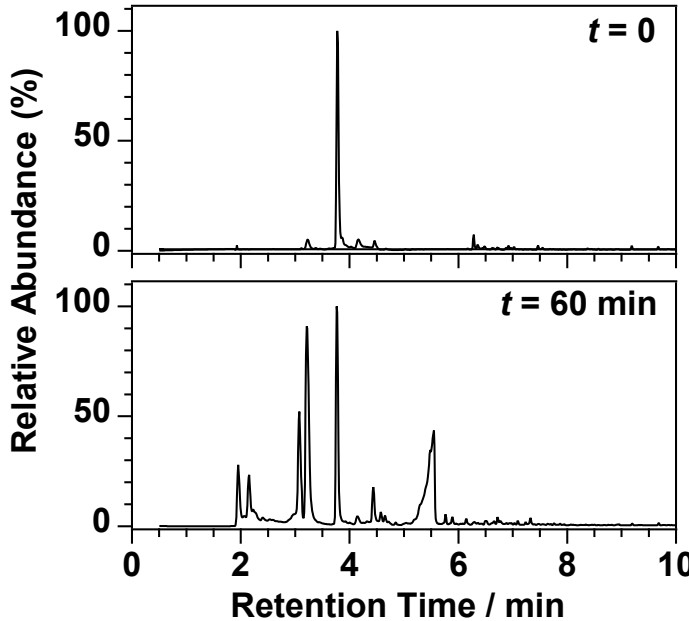

**Figure 6**. Chromatograms obtained for a 2MB/Cl$_2$ mixture before irradiation (top panel) and after 60 min of irradiation (bottom panel).





***Identification and quantification by FTIR Spectroscopy:*** **Figure 7** shows the FTIR spectrum recorded after 50 min of reaction time and the features of 2-methylbutanal subtracted. By comparison with the reference spectra (shown in **Fig. S6**), the most abundant products observed were HCl, butanone, and CO. The yield for butanone, $Y_{butanone}$, obtained from the plots shown in **Fig. S9a**, was (49.3±0.8)%. Formation of acetaldehyde and formaldehyde was also observed, but their quantification was very

imprecise due to the similarity between both IR spectra and the possible presence of other minor products such as propanal and 2-butanol. After subtracting HCl, butanone, and CO, ketene ($CH_2C(O)$) could be identified in the residual spectrum (shown in the bottom panel of **Fig. 7**) by comparison with the IR features reported by Wallington et al. (1996). The remaining bands could come from methylglyoxal, which was observed by GC-MS. It is worth noting that small amounts of acetaldehyde, formaldehyde, butanone and CO ($<9\times10^{12}$ molecules cm$^{-3}$) were observed during UV light exposure of 2-methylbutanal, but

their amount was negligible compared with the observed ones during the Cl reaction ($> 1 \times 10^{14}$ molecules cm$^{-3}$).

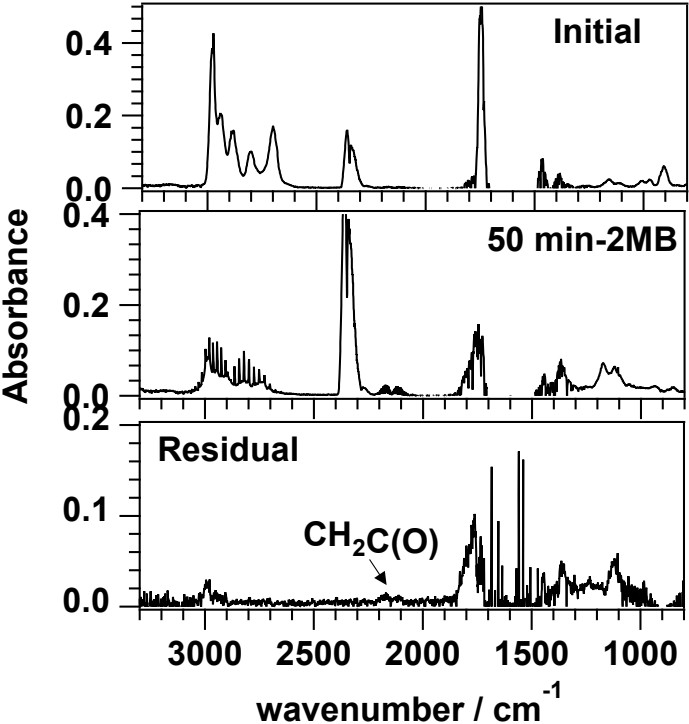

**Figure 7.** FTIR spectra used in the identification of the products in the Cl reaction of 2MB. The top panel shows the initial spectrum, the central panel shows the spectrum obtained after 50 min of Cl reaction, with the features of 2MB subtracted, and the bottom panel shows the residual spectrum after the subtraction of reaction products shown in the SI (**Figure S6**).



***Identification by PTR-ToF-MS:*** The identified reaction products, with an average ion concentration greater than or equal to 0.2 ppb, were acetaldehyde ($C_2H_4OH^+$, *m/z* = 45.03), butanone ($C_4H_8OH^+$, *m/z* = 73.06), methylglyoxal ($C_3H_4O_2H^+$, *m/z* = 73.03), 2-butanol ($C_4H_{10}OH^+$ *m/z* = 75.08), formaldehyde ($CH_2OH^+$ *m/z* = 31.02), ketene ($C_2H_2OH^+$, *m/z* = 43.02), methanol ($CH_4OH^+$, *m/z* = 33.03), 2-methylbutanoic acid ($C_5H_{10}O_2H^+$, *m/z* = 103.07), and propanal ($C_3H_6OH^+$, *m/z* = 59.05). All the products observed by GC-MS and FTIR, except CO and HCl, were also observed by PTR-ToF-MS. In the PTR-ToF-MS

analysis, it was observed that 2MB yields two different ions when ionized: $C_5H_{10}OH^+$ (39%) and $C_2H_4OH^+$ (61%). As the most abundant fragment from 2MB, $C_2H_4OH^+$, overlaps with the molecular ion from acetaldehyde, in order to quantify this product the contribution of 2MB to the $C_2H_4OH^+$ signal was eliminated taking into account its correlation with the $C_5H_{10}OH^+$signal when only 2MB was present in the chamber. In **Fig. 8** the time-evolution of 2MB (plotted as the sum of the two identified ions) and the main products, acetaldehyde, butanone, and methylglyoxal, are shown. The molar product yields, obtained from

the plots shown in **Fig. S9b**, were (58.4±0.2)%, (32.4±0.2)%, and (13.5±0.2)%, respectively. In addition, very low concentrations of 2-butanol were detected with a product yield of (0.11±0.03)%. It must be noted that formaldehyde and ketene are formed during UV light exposure of 2-methylbutanal in the test prior the Cl reaction, but at very low concentrations (<7.38×10$^{10}$ molecule cm$^{-3}$) compared with the observed after the Cl reaction.



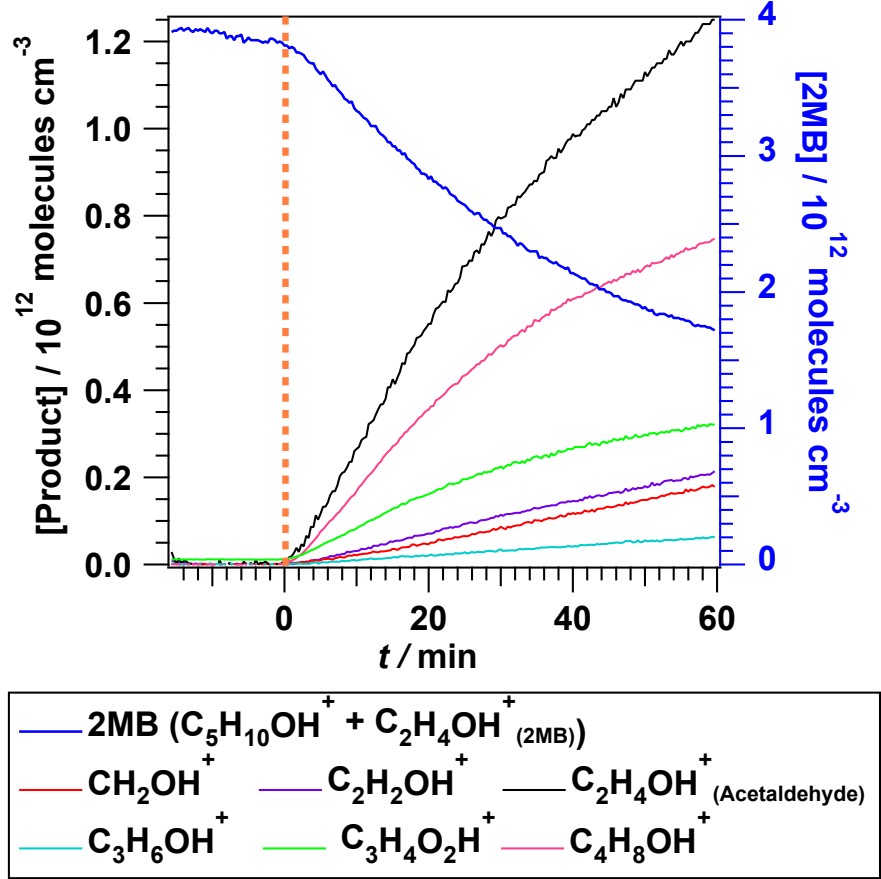

**Figure 8.** Temporal evolution of 2-methylbutanal (sum of the 2 identified ions) and the most abundant products measured by PTR-ToF-MS during the 2-methylbutanal + Cl reaction. Vertical dashed line indicates when UV lamps are switched on.

### 3.3.3 Formation of SOAs in the Cl + 2MB reaction

The size distribution of the particles formed in the Cl + 2MB reaction is shown in terms of the normalized particle number, $d\text{N}/d\log\text{Dp}$, and mass, $d\text{M}/d\log\text{Dp}$ in **Fig. S10**. After 2 min of reaction, particles of ca. 90 nm of diameter were formed and started to increase in concentration (particles cm$^{-3}$) and diameter as reaction elapsed. After 10 min, the particle number $d\text{N}/d\log\text{Dp}$ for a diameter of ca. 150 nm reached a maximum, while the maximum mass $d\text{M}/d\log\text{Dp}$ was reached at longer times ($t = 27$ min) and for particles of ca. 500 nm, what caused that the FMPS could not detect the total mass formed at times longer than 13 min in this case. Therefore, the data analysis to obtain $Y_{SOA}$ according to Eq. (3) was done only during the first 12-30 min, depending on the conditions. $Y_{SOA}$ determined under different conditions are listed in **Table 5**, ranging between 0.16 % and 0.76 %, and showing a negative dependence on $[2MB]_0/[Cl_2]_0$. It was observed an increase of $Y_{SOA}$ as the maximum





value of $M_{SOA}$ detected in the experiment increased. In previous works (Antiñolo et al., 2019; Antiñolo et al., 2020), this kind

of behaviour was described by the gas/particle absorption model proposed by Pankow (Pankow, 1994a, b) and fitted to the

equation proposed by Odum et al. (1996). However, in this work it was not possible to know the maximum $M_{SOA}$ formed in

the experiment due to instrumental limitations as observed in **Fig. S10b**, so data were not fit to this gas/particle absorption

model as in previous works.

**Table 5.** Results of $Y_{SOA}$ under different conditions.

| $[2MB]_0 / 10^{14}$ molecules $cm^{-3}$ | $[Cl_2]_0 / 10^{14}$ molecules $cm^{-3}$ | $[2MB]_0/[Cl_2]_0$ | $Y_{SOA}$ (%) |
|---|---|---|---|
| 6.9 | 10.3 | 0.67 | $0.62 \pm 0.04$ |
| 5.6 | 6.5 | 0.85 | $0.58 \pm 0.02$ |
| 7.6 | 8.9 | 0.86 | $0.76 \pm 0.02$ |
| 7.1 | 7.5 | 0.96 | $0.53 \pm 0.02$ |
| 7.4 | 5.4 | 1.4 | $0.66 \pm 0.02$ |
| 12 | 8.7 | 1.4 | $0.52 \pm 0.05$ |
| 7.2 | 4.8 | 1.5 | $0.30 \pm 0.01$ |
| 13 | 8.2 | 1.6 | $0.44 \pm 0.03$ |
| 12 | 6 | 2 | $0.16 \pm 0.02$ |

### 3.4 Photodegradation mechanisms

### 3.4.1 Mechanism of the UV photodissociation of 2MB

It is clear that the molecular elimination of CO (channel (R1c)) occurs because butane ($Y_{butane}$=9.8%) and CO were detected.

Although CO is directly produced in channel (R1c), it can also be rapidly produced by reaction of formyl radical, HCO, formed

in channel (R1b), with $O_2$ (**Figure S11b**). Butanone ($Y_{butanone}$=14.8%) is plausible to be formed from the radical forming

channels (R1a) and (R1b). The $CH_3CH_2CH(CH_3)CO$ and $CH_3CH_2CH(CH_3)$ radicals formed in those reactions, respectively,

react rapidly with $O_2$ to produce the corresponding peroxy ($RO_2$) radical that is involved in a sequence of reactions to generate

butanone, among other species. Other molecular photolysis pathways are the following ones (Gruver and Calvert, 1956;

Wenger, 2006):

$$CH_3CH_2CH(CH_3)C(O)H + h\upsilon \rightarrow CH_4 + CH_3CH=CHC(O)J \tag{R1d}$$

$$\rightarrow CH_2=CH_2 + CH_3CH=CHOH \tag{R1e}$$

None of these products were detected by PTR-ToF-MS or GC-MS. However, it is possible that $CH_3CH=CHC(O)H$ and/or

$CH_3CH=CHOH$, were responsible of the remaining bands observed in the residual IR spectrum although this could not be

checked due to the lack of reference spectra. Neither $CH_4$ nor $CH_2=CH_2$ could be clearly detected.


### 3.4.2 Mechanism of the 2MB + Cl reaction

After evaluating the gas-phase products formed in the Cl-reaction of 2-methylbutanal, some information can be inferred concerning the reaction mechanism. The presence of HCl as a primary product and the fact that no other chlorinated product

was observed indicate that the reaction proceeds through the H-abstraction from different sites in 2MB. There are five susceptible reaction sites in 2MB:

$$Cl + CH_3CH_2CH(CH_3)C(O)H \rightarrow CH_3CH_2CH(CH_3)\mathbf{C(O)} + HCl \qquad (R3a)$$
$$\rightarrow CH_3CH_2\mathbf{C}(CH_3)C(O)H + HCl \qquad (R3b)$$
$$\rightarrow CH_3\mathbf{CH}CH(CH_3)C(O)H + HCl \qquad (R3c)$$

$$\rightarrow \mathbf{CH_2}CH_2CH(CH_3)C(O)H + HCl \qquad (R3d)$$
$$\rightarrow CH_3CH_2CH(\mathbf{CH_2})C(O)H + HCl \qquad (R3e)$$

The mechanism for the H-abstraction from the HCO group is depicted in **Fig. S11a**. The possible H-abstraction channels from the hydrocarbon chain are depicted in **Fig. S12.** As shown in the figures, acetaldehyde ($CH_3C(O)H$), formaldehyde ($HC(O)H$)

or methanol ($CH_3OH$) can be explained by any of the 5 possibilities, $CH_3CH_2C(O)CH_3$ can be only formed if the aldehydic H (**Figure S11a**) or the tertiary H in C-2 are abstracted (**Figure S12b**), and methylglyoxal ($CH_3C(O)C(O)H$) is a product only when a hydrogen atom from C-2, C-3, or C-4 is abstracted (**Figure S12b-d**). The product yields obtained in this work indicate that the channels for the H-abstraction from the -C(O)H group and the tertiary H in C-2 are more favoured, given that $Y_{butanone}$ is more than 2 times higher than $Y_{methylglyoxal}$.


### 4. Atmospheric implications

Considering the most important diurnal degradation pathways (UV photolysis, reaction with OH radicals and Cl atoms), the tropospheric lifetime of 2MB, $\tau$, can be estimated according to Eq. (7). To compare the relative importance of these three degradation routes on $\tau$, each term was estimated for a coastal city (Valencia, Spain) at sea level ($z=0$) as a function of time

from 6:00 to 22:00 LT (local time, corresponding to GMT+2).

$$\frac{1}{\tau} = \frac{1}{J(z,\theta)} + \frac{1}{k_{OH}[OH]} + \frac{1}{k_{Cl}[Cl]} \qquad (7)$$

Therefore, in Eq. (7) $J(z,\theta)$ is the calculated photolysis rate of 2MB at sea level and with a zenith solar angle ($\theta$) varying from 16º at 14:00 LT and 96º at 6:00 LT. $k_{OH}$ and $k_{Cl}$ are those determined in this work at 298 K, and [OH] and [Cl] are tropospheric concentrations of OH and Cl. Temporal values of [OH] are considered to be similar to those modelled by Forberich et al.

(1999) for a day at the end of June at Weybourne (UK), location with similar $\theta$ than Valencia in June, except for the central time of the day. [Cl] was considered as a time-independent upper limit value and taken as the peak concentration, $1.3 \times 10^5$ atoms cm$^{-3}$, predicted by Spicer et al. (1998) in marine environments.





The photolysis rate of 2MB $J(z,\theta)$ is defined as follows (Jiménez et al., 2007):

$$J(z,\theta) \cong \Phi_{eff} \sum_{\lambda>290nm} F(\lambda,z,\theta)\sigma_\lambda \Delta\lambda \tag{8}$$

where $F(\lambda,z,\theta)$ (in photons cm$^{-2}$ nm$^{-1}$ s$^{-1}$) is the solar spectral actinic flux at $z$ for a specific $\theta$ in the troposphere, obtained using

the TUV radiative transfer model (5.3 version) developed by Madronich and Flocke (1999) and $\Delta\lambda = 1$ nm. $F(\lambda,z,\theta)$ was set

for the summer solstice at sea level ($z = 0$ km) in a coastal city in Spain, Valencia, varying $\theta$ between 16º and 96º, as explained

above. $\sigma_\lambda$ used in the calculation were those listed in **Table S1** and $\Phi_{eff} = 0.30$, determined in this work. The estimated $J(z,\theta)$

values at sea level as a function of the zenith angle are provided in **Table 6**. As expected, it is observed that photolysis is faster

at 14:00 ($J(z,\theta) = 2.90\times10^{-5}$ s$^{-1}$) when the solar actinic flux is maximum, whereas it is negligible at the beginning and the end

of the day.

Table 6. Estimated photolysis rate coefficients of 2-methylbutanal at sea level ($z = 0$ km) in the summer solstice day in Valencia (Spain).


| Local time (GMT+2) | $\theta$ / º | $J(z,\theta)$/ $10^{-5}$ s$^{-1}$ |
|---|---|---|
| 6:00 | 96 | ~ 0 |
| 8:00 | 76 | 0.356 |
| 10:00 | 53 | 1.51 |
| 12:00 | 31 | 2.51 |
| 14:00 | 16 | 2.90 |
| 16:00 | 29 | 2.55 |
| 18:00 | 52 | 1.57 |
| 20:00 | 75 | 0.406 |
| 22:00 | 95 | ~ 0 |

At dawn the estimated $\tau$ is 8 h, whereas at 14:00 LT it is 1 h. The relative contribution of the degradation routes evaluated in

this study to the diurnal loss of 2MB is depicted in **Fig. 9**. At dawn and twilight, the Cl-reaction dominates the loss of 2MB

with a relative contribution of 84%, followed by the OH-reaction (16%). On the other hand, in the central times of the day,

OH-reaction is clearly the main removal route for 2MB, with a relative contribution of 79% at 14:00, followed by a competition

between photolysis (11%) and Cl-reaction (10%). Note that the relative contribution of Cl-reaction is an upper limit as the

considered [Cl] is a peak value.





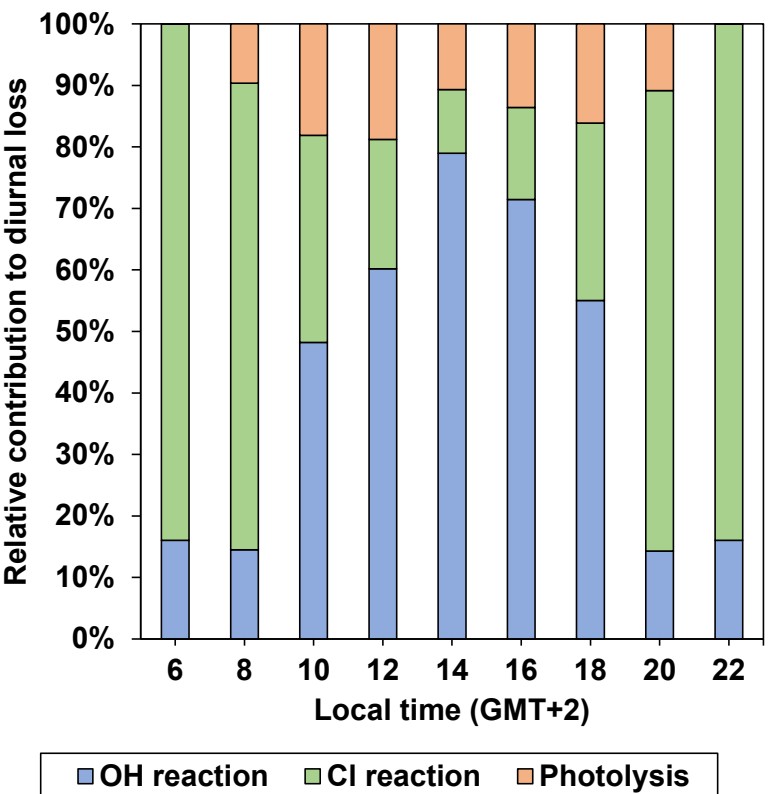

**Figure 9**. Relative contribution of the three removal routes studied in this work to the total diurnal loss.

As stated above, once emitted, 2MB is degraded in few hours during daytime, so it will not be transported to long distances. Its degradation products include CO and carbonyl compounds such as butanone, acetaldehyde, or methylglyoxal, that can be further oxidized in the troposphere and contribute to photochemical smog, impacting on human health. In terms of SOA formed, the evidence found in this work showed that few particles are formed in the Cl reaction ($Y_{SOA}$ = 0.16 - 0.76 %). Finally, HCl, detected in the oxidation by Cl, can contribute to acid rain. However, it must be noted that the impact of the observed products has a strong dependence on the amounts of 2MB emitted to the troposphere.





## Author contribution

M. As., M. An., and S. B. designed and conducted the experiments, and analysed the experimental data. E.J. and J.A. designed and supervised the experiments and managed the project. All the co-authors have contributed to prepare the manuscript and to
discuss the obtained results.

## Acknowledgements

This work has been supported by the Regional government of Castilla-La Mancha through CINEMOL project (Ref.: SBPLY/19/180501/000052) and by the University of Castilla-La Mancha – UCLM (*Ayudas para la financiación de*
*actividades de investigación dirigidas a grupos* (REF: 2019-GRIN-27175). M. Asensio, M. Antiñolo and S. Blázquez also acknowledge UCLM (*Plan Propio de Investigación*) for funding their contracts during the performance of this investigation.

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
