# Peer review of "Evaluation of the daytime tropospheric loss of 2-methylbutanal"

_Atmospheric Chemistry and Physics, 2021_

## Referee Comment (RC1)

Manuscript ID: acp-2021-758

Authors: María Asensio, María Antiñolo, Sergio Blázquez, José Albaladejo, Elena Jiménez

Ms. Title: Evaluation of the daytime tropospheric loss of 2-methylbutanal.

**General Comments:**

In the present work, the authors have thoroughly and comprehensively studied the primary tropospheric losses of 2-methylbutanal (2MB) performing precise OH (PLP-LIF method) and Cl (Relative Rate method) kinetic measurements, as well as determining the photolysis frequency, *J*, and the effective quantum yield,  $\Phi_{eff}$ , at actinic wavelengths. As part of this work, the authors have measured the UV absorption cross-section (220 – 360 nm), they have determined Cl reaction products using GC-MS, FTIR and PTR-TOF-MS as detection techniques and they have looked into the SOA formation contribution of Cl chemistry. Using the data obtained in this work the authors have estimated the lifetime of 2MB and the daytime contribution of the three different loss processes, at Valencia (city of Spain) environment, as case study. Therefore, the present reviewer believes that the content of the present work meets the high standards of ACP. However, there are some presentation issues and a few scientific comments that authors need to address before the manuscript will be in a publishable form. Generic and specific suggestions and comments that the present author believes they will improve both quality and impact of the manuscript are given below.

**Suggestion: Published after minor corrections**

**Specific Comments:**

1. In the abstract, the authors have a typo on the rate coefficient for Cl atoms reaction with 2MB. The correct order should be  $10^{-10}$ , i.e.,  $k(298 \text{ K}, 760 \text{ Torr}) = (2.16 \pm 0.16) \times 10^{-10} \text{ cm}^3$  molecule-1 s-1.

2. Although tropospheric chemistry (OH and Cl) and photolysis are expected to dominate the atmospheric degradation of 2MB, deposition should not be neglected, especially in the case that a unique product might be formed during wet or dry deposition followed by on surface hydrolysis. It might be worth the authors to address that in the introduction.

3. The authors should give in the text the magnitude of the total corrections (in the whole course of the measurement, e.g., total reaction or photolysis time) has been made, in kinetic and photolysis measurements, due to other than the primary processes of interest. This way the reader can evaluate the accuracy of the measurements and the potent errors that these corrections introduce.

4. As expected, OH chemistry dominates the lifetime of 2MB and thus it would have been of interest to determine OH +2MB reaction products in a chamber experiment, since PLP-LIF method would not allow that. Were there any limitations that did not allowed the authors to look into OH initiated chemistry end-products determinations?

5. Although the authors clearly report how they have estimated 2MB lifetime at Valencia atmosphere, the use of a peak Cl-concentration and treating that as a constant diurnally is deceptive, particularly since after dawn, Cl concentration is expected to be rapidly decreased. The constant [Cl] treatment leads to a unexpectedly high impact of Cl chemistry on 2MB loss especially after 18:00. It might be worth the authors to revisit their analysis and try to address that.

6. A conclusion section in which the conclusive findings of this work would be summarized is entirely missing from the manuscript and could be extremely helpful for the communication between the authors and the reader to evaluate this nice piece of work, particularly since there is quite extensive information. It is strongly recommended to be added.

**Technical Corrections:**

**1. Abstract, line 14**: Please correct the order of magnitude for  $k_{Cl}$  and add "respectively", at the end of the sentence, e. g.,  $(2.16\pm0.16)\times10^{-10}$  cm3 molecule-1 s-1, respectively.

**2. Pg. 2, line 45**: Please replace "clean atmosphere" with "free-atmosphere" or even better with "NOx-free atmosphere".

3. Pg 3. line 81: Please add reference for U.S. Standard Atmosphere

4. **Pg 5. line 127**: Please replace "expression" with "decay".

5. **Pg 5. line 140**: Please replace "different detectors" with "a variety of detection techniques".

6. **Pg 6. line 151**: Please replace "Relative kinetics" with "Relative rate kinetics"

7. **Pg 7. line 183**: Please replace "*during 60 min*" with "*for 60 min*".

8. **Pg 7. Line 184**: Please replace "*in the 2MB*" with "*of 2MB*".

**9. Pg 7. line 187**: The phrase "*with an air flow by means a dynamic inlet...*" is not clear to the present reviewer. Please revise.

**10. Pg 8. line 211, 213 and 214**: Please delete "...*all*...", "...*of the present as*..." and "...*is*...", respectively.

11. Pg 8. line 224-225: Please replace "...unknown – this work." with "...unknown. It is worth to note though that in the same study, that refers to structurally similar aldehydes, e. g., pentanal or 3-methyl-butanal, the reported  $\Phi_{eff}$ , (0.30±0.02) and (0.27±0.01), respectively, were closer to the one determined in this work".

**12. Pg 10 line 242**: Please replace "*is the molecular one yielding*" with "*yields close-shell molecules, i.e.,* …".

13. Pg 10. lines 249 – 251: Please rephrase the whole sentence to avoid confusion.

**14. Pg 10. lines 247 and 248**: Please replace "*as*" with "*from*" and "*were occurring*" with "*was present*".

**15. Pg 11. lines 266-267**: Please replace "...*compound, such an aldehyde or ketone.*" with "...*group, alluding to aldehydes or ketones formation.*".

**16. Pg 11. line 269**: *"Therefore, ...products"* is a very generic statement. It is suggested to just say that have not been assigned to other atmospheric oxidation products.

17. Pg 12. line 274-275: Please rephrase to avoid confusion

**18. Pg 12. line 285**: Please replace "*k*OH(T=263-353 K)" with "*k*OH(T)"

**19. Pg 12. lines 287 and 288**: Please replace "*the obtained*" with "*the one obtained*" and "*relative kinetic*" with "*relative rate kinetic*", respectively.

**20. Pg 12. lines 290 and 292**: Please replace "*They used only*" with "*They only used*" and "..., *could have suffered...products.*" with "*might interfere with reaction products bands*", respectively.

21. Pg 15. Table 4: The quoted error limit in the average does not seem correct or the result from error propagation. If you include the extremes of the *k* measured based on the two reference reactions, k with error limits should be:  $(2.16 \pm 0.32) \times 10^{-10}$  cm3 molecule-1 s-1, in the case you don't want to use asymmetric error limits.

**22. Pg 16. line 331**: Please replace "time and features...subtracted." with "time. The features of 2MB have been subtracted for clarity purposes".

**23. Pg 16. lines 334-336**: It should have been possible to discriminate formaldehyde from acetaldehyde pretty easily. The features are rather discrete and IR spectra readily available. Please elucidate.

**24. Pg 16. Figure 7**: Please capitalize w in Wavenumber and check the displacement of CH2...C(O), in the bottom panel.

**25. Pg 17. line 363**: Please replace "*with the observed after*" with "*the observed levels, after*".

26. Pg 18-19. lines 378-381: Rephrase to avoid confusion.

**27. Pg 19. line 382-383**: Please replace "...as observed.... previous works." with "as depicted in Fig S10b, and thus our data were not fitted with gas/particle adsorption model.".

28. Pg 19. line 389: Please replace "because" with "since"

29. Pg 19. line 394: Please delete "the following ones"

**30. Pg 19. line 401**: Please replace "*checked*" with "*justified*".

**31. Pg 20. line 403 and 405**: Please replace "of" with "with", "product was" with "products were" and "through the" with "via".

**32. Pg 20. line 415 and 429**: Please replace "(*CH*3*OH*)" with "(*CH*3*OH*) *formation*" and "*OH and Cl*" with "*OH radicals and Cl atoms*".

**33. Graphs:** In all graphs it would be easier for the reader to see the units in parenthesis without math signs, e. g., instead of Wavenumber  $/cm^{-1}$ , use Wavenumber  $(cm^{-1})$ .

---

## Author Response (AR1)

**REPLY TO REFEREE #1**

We greatly appreciate the Referee #1's comments on our work that certainly help us to improve the quality of the submitted manuscript. In the following sentences we reply to the comments.

**Reply to Specific Comments:**

*1. In the abstract, the authors have a typo on the rate coefficient for Cl atoms reaction with 2MB. The correct order should be $10^{-10}$, i.e., k(298 K, 760 Torr) = (2.16 $\pm$ 0.16) $\times 10^{-10}$ cm³ molecule⁻¹ s⁻¹.*

> **Authors' reply:** Yes, it was a typo. We will correct it in the revised version of the manuscript that will be uploaded after the discussion is finished.

*2. Although tropospheric chemistry (OH and Cl) and photolysis are expected to dominate the atmospheric degradation of 2MB, deposition should not be neglected, especially in the case that a unique product might be formed during wet or dry deposition followed by on surface hydrolysis. It might be worth the authors to address that in the introduction.*

> **Authors' reply:** As far as we know, there is a lack of information on the dry and wet deposition of 2-methylbutanal. However, following the Reviewer's advice, we have included a sentence in the introduction to point out the possibility of deposition as a sink for this compound, similarly to acetaldehyde, for which deposition is a minor sink (Millet et al.; Atmos. Chem. Phys., 10, 3405–3425, 2010). In fact, it is possible to estimate the lifetime of 2MB when only wet deposition is considered ($\tau_{Wet}$) since its Henry's Law constant is known ($k_{H,cp}$ =2.33 M atm⁻¹ (Pollien et al.; Int. J. Mass Spectrom., 228, 69-80, 2003)). $\tau_{Wet}$ can be estimated from the following equation (Chen et al.; Atmos. Environ., 37, 4817-4822, 2003):
>
> $$\tau_{Wet} = \frac{z}{v_p R T k_{H,cp}}$$
>
> where $z$ is the altitude, $v_p$ is the precipitation rate, that can be considered as 0.43 m year⁻¹, which is the annual average precipitation in Valencia (Spain), R is the gas constant and $T$ is temperature, that can be considered to be 298 K. Taking all this into account, and considering different altitudes between 100 and 2000 m, $\tau_{Wet}$ is estimated to range between 4 and 82 years, depending on the considered altitude. If this estimation is compared with the tropospheric sinks discussed in the original manuscript, it is clear that wet deposition is a negligible sink for this compound.

*3. The authors should give in the text the magnitude of the total corrections (in the whole course of the measurement, e.g., total reaction or photolysis time) has been made, in kinetic and photolysis measurements, due to other than the primary processes of interest.*

*This way the reader can evaluate the accuracy of the measurements and the potent errors that these corrections introduce.*

**Authors' reply:** The correction made in the Cl-reaction kinetic experiments, as described in the manuscript, includes the overall losses for both 2MB and the reference compound (heterogeneous reaction onto the reactor walls, UV photolysis and/or reaction with the oxidant precursor). If this correction had not been applied, the rate coefficient would have been $(2.07 \pm 0.31) \times 10^{-10}$ cm$^3$ molecule$^{-1}$ s$^{-1}$, which shows a 4% difference with the corrected result: $(2.16 \pm 0.32) \times 10^{-10}$ cm$^3$ molecule$^{-1}$ s$^{-1}$. The contribution of the mentioned losses will be included in the revised manuscript.

For the photolysis experiments, only the loss onto the walls of 2MB was considered. If no correction had been done, the result would have been $(2.57 \pm 0.15) \times 10^{-5}$ s$^{-1}$ instead of $(1.96 \pm 0.32) \times 10^{-5}$ s$^{-1}$, so they are 24% different. The magnitude of this correction was already mentioned in Section 2.1.2, and in the revised version that will be pointed out when presenting the results.

*4. As expected, OH chemistry dominates the lifetime of 2MB and thus it would have been of interest to determine OH +2MB reaction products in a chamber experiment, since PLP-LIF method would not allow that. Were there any limitations that did not allowed the authors to look into OH initiated chemistry end-products determinations?*

**Authors' reply:** As pointed out by the Reviewer, the PLP-LIF experimental set-up used in the kinetic study of the OH-reaction does not allow to determine end-products of the reaction and a chamber experiment would be useful. However, to detect reaction products from OH+2MB reaction in our chambers some experimental complications arise. The chamber coupled to the FTIR spectrometer is made of Pyrex, not quartz, which forces us to use actinic lamps ($\lambda_{max} = 360$ nm). At this wavelength, CH$_3$ONO/O$_2$/NO should have been used as the OH-precursor. As NO is needed to convert HO$_2$ radical into OH, its presence would change the fate of the radicals formed in the OH+2MB reaction and different reaction products than those formed in the absence of NO would be formed, making harder to interpret the FTIR spectra because of the overlap of the IR bands. To get cleaner reaction mixture, photolysis of H$_2$O$_2$ by germicide lamps ($\lambda_{max} = 254$ nm) should be used to generate OH radicals. These lamps can only be used with the 264-L chamber, that has quartz windows, but photolysis of both 2MB and reaction products can also occur complicating the FTIR spectrum after irradiation with new bands from photolysis products or increasing the intensity of the IR bands that are common to reactive and photolysis processes. Monitoring the Cl-reaction products is usually easier than for the OH-reactions since the actinic lamps ($\lambda_{max} = 360$ nm), used to photolyze Cl$_2$, minimize the photolysis of 2MB and reaction products. As measured in this work, the absorption cross section of 2MB at 360 nm is much lower, $(0.03 \pm 0.08) \times 10^{-20}$ cm$^2$, than that at 254 nm, $(1.01 \pm 0.09) \times 10^{-20}$ cm$^2$. Moreover, Cl atoms are less selective than OH radicals, so the Cl-reaction usually yields the same products than OH-reaction and some other products too. Then, we

decided that the information obtained in the study of the products of the Cl-reaction should be enough to understand the atmospheric implications of 2-methylbutanal. In fact, there are previous works that look at the Cl-reaction products to understand the products of the OH-reaction, such as the study of the unsaturated aldehydes published by Orlando and Tyndall (*J. Phys. Chem. A* **2002**, *106*, 12252-12259).

*5. Although the authors clearly report how they have estimated 2MB lifetime at Valencia atmosphere, the use of a peak Cl-concentration and treating that as a constant diurnally is deceptive, particularly since after dawn, Cl concentration is expected to be rapidly decreased. The constant [Cl] treatment leads to an unexpectedly high impact of Cl chemistry on 2MB loss especially after 18:00. It might be worth the authors to revisit their analysis and try to address that.*

**Authors' reply:** We agree with the Reviewer that the effect of Cl atoms on the total loss of 2MB is overestimated in our work since we considered a constant peak concentration, $[Cl]_{peak}$, of $10^5$ cm$^{-3}$. For the first hours of the day, $[Cl]_{peak}$ could probably work fine, but the Referee is right when pointing out that the impact is surprisingly (and erroneously) high after 18:00. As far as we know, there is no hourly measurements of [Cl] in Valencia (Spain) or a similar location. That makes hard to provide an accurate hourly contribution of Cl atoms to the loss of 2MB. From 20:00 h, obviously solar radiation, that can generate both Cl atoms and OH radicals, is less intense. Therefore, we will remove the relative contributions of OH, Cl, photolysis at 20:00 and 22:00 in Figure 9. Anyways, we will keep them for the rest of the day, taking into account that, even using $[Cl]_{peak}$, the major tropospheric route for the homogeneous loss of 2MB is the OH-reaction. Figure 9 will be replaced by this one:

[Figure]

**Figure 9**. Relative contribution of the three removal routes studied in this work to the total diurnal loss.

Following the Referee 2's suggestion, we will also include in the revised manuscript the estimation of the tropospheric lifetime considering the average concentrations in 24 h of OH and Cl, and averaged values of the solar spectral actinic flux for 24 h in Valencia. The following text will be included:

"Considering 24 h-average values of [OH] ($1 \times 10^6$ radicals cm$^{-3}$, Krol et al., (1998)), [Cl] ($1 \times 10^3$ atoms cm$^{-3}$, Singh et at. (1996)), and $F(\lambda, z, \theta)$ for Valencia in June, $\tau$ can be estimated as 7.5 h for a 24 h-period, with a relative contribution of 73% of the OH-reaction, 27% of photolysis, and less than 1% of the Cl-reaction."

*6. A conclusion section in which the conclusive findings of this work would be summarized is entirely missing from the manuscript and could be extremely helpful for the communication between the authors and the reader to evaluate this nice piece of work, particularly since there is quite extensive information. It is strongly recommended to be added.*

**Authors' reply:** We really thank the Reviewer for this comment. Although some conclusions were included at the end of the "Atmospheric Implications" section, we will certainly add a "Conclusions" section in the revised version of the manuscript to summarize the findings of this work:

**5. Conclusions**

This work presents the relative importance of the most important diurnal atmospheric degradation routes of 2MB and the formation of secondary pollutants (particulate matter and gaseous products). In particular, it presents the study of the UV photolysis kinetics at $(298 \pm 2)$ K and $(760 \pm 3)$ Torr; it describes, for the first time, the temperature dependence of the OH reaction between 263 and 353 K and at P = 50-600 Torr of He; and it reports, for the first time, the rate coefficient of the Cl reaction at 298 K and 760 Torr. It was found that OH-reaction is the main removal route for 2MB at central times of the day, with a relative contribution of 79% at 14:00 LT on a day of June in Valencia (Spain) and a tropospheric lifetime of 1 h.

In terms of the degradation products, butanone, acetaldehyde, or methylglyoxal were detected as major products in the Cl-reaction, together with CO and HCl. During photolysis, butane, butanone, and CO were among the products identified. SOA formation was detected, although at yields lower than 0.76%. The carbonyl products formed can contribute to photochemical smog, impacting on human health, whereas HCl can contribute to acid rain, the real impact will depend on the amounts of 2MB emitted to the troposphere.

**Reply to Technical Corrections**

All small changes suggested by the Reviewer will be addressed in the revised version of the manuscript and not listed here.

*3. Pg 3. line 81: Please add reference for U.S. Standard Atmosphere.*

> **Authors' reply:** In the revised manuscript, the reference for U.S. Standard Atmosphere will be included:
>
> Gueymard, C. A., Myers, D., and Emery, K.: Proposed reference irradiance spectra for solar energy systems testing, Solar Energy, 73, 443-467, https://doi.org/10.1016/S0038-092X(03)00005-7, 2002.

*9. Pg 7. line 187: The phrase "with an air flow by means a dynamic inlet…" is not clear to the present reviewer. Please revise.*

> **Authors' reply:** There was a typo in this sentence. In the revised manuscript it will state: "with an air flow by means of a dynamic inlet dilution system".
>
> The "dynamic inlet dilution system" is an add-on of the PTR-ToF-MS instrument that is used to dilute the sample gas flow by a clean gas flow to lower the sample gas concentration in a defined way. This is necessary when the concentration of a sample compound exceeds several ppm, which negatively affects the quantification. The sample gas flow is diluted by adding a defined clean gas flow, mixing with the sample, and then subtracting a different defined flow from the mixture, so a lower flow is introduced in the reaction chamber. This approach has the advantage that the sample gas does not need to pass through any components, such as flow controllers, which could impact the trace concentrations through surface interactions, etc.

*13. Pg 10. lines 249 – 251: Please rephrase the whole sentence to avoid confusion.*

**Authors' reply:** The sentence will be rephrased as: "In addition, $Y_{CO}$ was observed to increase when the initial concentration of 2MB decreased. On the other hand, butane was quantified…".

*16. Pg 11. line 269: "Therefore, ...products" is a very generic statement. It is suggested to just say that have not been assigned to other atmospheric oxidation products.*

**Authors' reply:** The sentence will be rephrased in the revised manuscript as: "However, the remaining IR bands in the residual spectrum could not be assigned to other gaseous oxidation products."

*17. Pg 12. line 274-275: Please rephrase to avoid confusion.*

**Authors' reply:** The sentence will be rephrased in the revised manuscript as: "For that reason, after 150 min of photolysis, the maximum photolysis time used in this work, the content of the photolysis cell was transferred to the 16-L chamber and diluted in synthetic air. This diluted sample was flowed through the PTR-Tof-MS during 5 min, in which mass spectra were measured. From the averaged mass spectrum, the products formed after 150 min of irradiation were identified. Therefore, no temporal evolution of products was measured."

*18. Pg 12. line 285: Please replace "$k_{OH}(T=263\text{-}353\ K)$" with "$k_{OH}(T)$".*

**Authors' reply:** We decided to add the temperature range to emphasize that the shown expression is valid only in that range. We believe that it is better to show it in this way to make it clear to the reader; however, we will replace $k_{OH}$(T=263-353 K) by $k_{OH}$(263-353 K) for simplicity.

*21. Pg 15. Table 4: The quoted error limit in the average does not seem correct or the result from error propagation. If you include the extremes of the k measured based on the two reference reactions, k with error limits should be: $(2.16 \pm 0.32) \times 10^{-10}\ cm^3\ molecule^{-1}\ s^{-1}$, in the case you don't want to use asymmetric error limits.*

**Authors' reply:** We thank the Referee for pointing out that the error limits in Table 4 were not correct. We stated in the beginning of the manuscript that errors were quoted as $\pm 2\sigma$, but the table contained them as $\pm \sigma$. We will correct Table 4 in the revised manuscript so the errors in $k_{Cl}$ are $\pm 2\sigma$:

**Table 4.** Results obtained in the kinetic experiments for the gas-phase reaction of Cl with 2-methylbutanal at $298 \pm 2$ K and $760 \pm 5$ Torr of air.

| Reference | $k_{Cl}/k_{Ref}$ | $k_{Ref}$ / $10^{-10}$ cm$^3$ molecule$^{-1}$ s$^{-1}$ | $k_{Cl}$ / $10^{-10}$ cm$^3$ molecule$^{-1}$ s$^{-1}$ |
|---|---|---|---|
| Ethanol | $2.140 \pm 0.038$ | $1.00 \pm 0.06$ [1] | $2.14 \pm 0.27$ |
| Isoprene | $0.509 \pm 0.003$ | $4.30 \pm 0.58$ [2] | $2.19 \pm 0.59$ |
| **Average** | | | $\mathbf{2.16 \pm 0.32}$ |

[1] Atkinson et al. (2006), [2] Orlando et al. (2003)

*23. Pg 16. lines 334-336: It should have been possible to discriminate formaldehyde from acetaldehyde pretty easily. The features are rather discrete and IR spectra readily available. Please elucidate.*

**Authors' reply:** We agree with the Referee. The IR spectrum of HC(O)H and CH$_3$C(O)H can be easily differentiated; however, at the very low absorptions observed in the residual spectrum, the quantification of these compounds by spectral subtraction was neither accurate nor reproducible. In addition, given that formaldehyde and acetaldehyde share the position of some bands (for example, the C=O stretching at 1750 cm$^{-1}$) and other products such as propanal and 2-butanol can also interfere at some other bands, we decided not to present these inaccurate results in the manuscript. In the revised version of the manuscript, the problems found in the quantification of HC(O)H and CH$_3$C(O)H will be pointed out.

*26. Pg 18-19. lines 378-381: Rephrase to avoid confusion.*

**Authors' reply:** The sentence will be modified in the revised manuscript as: "$Y_{SOA}$ was observed to increase as the maximum value of M$_{SOA}$ detected in the experiment increased. In previous works (Antiñolo et al., 2019; Antiñolo et al., 2020), this trend was described by the gas/particle absorption model proposed by Pankow (1994a, b) and fitted to the equation proposed by Odum et al. (1996)."

*33. Graphs: In all graphs it would be easier for the reader to see the units in parenthesis without math signs, e. g., instead of Wavenumber /cm$^{-1}$, use Wavenumber (cm$^{-1}$).*

**Authors' reply:** We agree with the Reviewer that it can be easier for readers if units are in brackets, for example, in a graph like the shown in Figure 5 in which it is hard to say what the magnitude is and what the units are, but IUPAC recommends to note the units as we presented them in our tables and graphs (see the manual *Quantities, Units and Symbols in Physical Chemistry*) and we believe that we should follow the IUPAC recommendation.

**REPLY TO REFEREE #2**

We thank the Reviewer for the comments and suggestions made that will increase the quality of this work.

*Overall, I find the UV absorption cross-sections and kinetic OH and Cl kinetic rate coefficients to be of high quality, useful for atmospheric research and appropriate for publication in the ACP journal. However, the product study analysis and reporting of SOA formation I find to be less useful, and in need of additional refinement if they are to be included in the main document. This manuscript reports a tremendous amount of work. However, more is not always better, and the paper would increase its impact potential if itwere focus more on the results with higher certainty.*

> **Authors' reply:** As we discussed in the replies below, the purpose of the product study was to see if the degradation of 2MB initiated by reaction with Cl and by UV solar radiation may form secondary pollutants even more harmful that the primary one. The yield of SOA formed in the Cl-reaction of 2MB is too small (less than 0.8%) to potentially affect the human health. For that reason and following the Reviewer's suggestion, we will move section 3.3.3 to the SI in the revised version of the manuscript. In terms of the gas-phase products, under the experimental conditions of our smog chamber experiments, the present results may not be generally applicable to the real atmosphere, as pointed out by the Reviewer, since the $RO_2+RO_2$ reactions are expected to be more important in our work than $RO_2+HO_2$ chemistry which dominate in non-polluted atmospheres. Nevertheless, the end-products in a real atmosphere coming from $HO_2$ chemistry would be the same as those observed in our experiments. For that reason, the discussion on the degradation mechanism of 2MB was kept short in the original manuscript and the detailed mechanism was included in the SI. We also agree to include more information about the $HO_2$ chemistry in the revised SI and will revise the conclusions of this work to make clear that our conditions are not the same as in a real atmosphere.

*Photolysis mechanism: more focus on this aspect could strengthen the paper. Simple calculations of the energetics of the possible decomposition pathways for atmospherically relevant photons may be helpful. Providing additional information to support the branching ratio constraints would be helpful.*

**Authors' reply:** With the collaboration of Prof. Lucía Santos (University of Castilla-La Mancha), we have performed calculations at the BHandLYP/ 6-311++G** level, similarly to what Castañeda et al. (J. Mex. Chem. Soc. 2012, 56(3), 316-324) did for aliphatic aldehydes, to obtain $\Delta G^0$ at 298 K and the $\lambda_{threshold}$ for the three photolysis channels of 2MB.

The obtained results are summarized in the following table:

| Channel | Photoproducts | $\Delta G^0_{298\ K}$ / kJ mol$^{-1}$ | $\lambda_{threshold}$ / nm |
|---------|---------------|-----------------------------------------|------------------------------|
| 1a | $CH_3CH_2CH(CH_3)C(O) + H$ | 326.1 | 367.1 |
| 1b | $CH_3CH_2CH(CH_3) + HC(O)$ | 358.3 | 334.1 |
| 1c | $CH_3CH_2CH_2CH_3 + CO$ | -53.2 | |

From the energetics of these reactions, it can be inferred that all channels are feasible in the troposphere, where the available UV radiation has a minimum wavelength of 290nm. From the thermodynamic point of view, the branching ratios cannot be calculated since kinetics plays a crucial role. Kinetic calculations, although possible, require more computational time than the thermodynamic calculations presented here and are out of the scope of this paper.

*For example, the band used for quantification of butane (C-H stretch) overlaps highly with the precursor and likely, many products. Given this, how well is butane actually quantified? Is butane quantified by other methods?*

**Authors' reply:** We agree with the Referee that it is hard to directly obtain reliable results when several IR bands overlap in the spectrum. To minimize that interference, the subtraction procedure of the final spectrum was:

1) Subtraction of the IR features from the unreacted 2MB, focusing on the 2700 cm$^{-1}$ and 2800 cm$^{-1}$ bands.

2) Once 2MB was subtracted, the IR features of butanone were subtracted, focusing on the C=O stretching band at 1750 cm⁻¹.

3) Finally, the IR features of butane were subtracted of the resulting spectrum, focusing on the 2800-3000 cm⁻¹ band.

In all steps, we always looked to the whole spectrum, not only to the selected bands, to be sure that the subtraction was properly done. So, the subtraction procedure led us confidence in the quantification of butane by FTIR spectroscopy. We will clarify this in revised manuscript.

Butane could not be detected by GC-MS because it was not absorbed on the SMPE fibre that was used to collect the sample. It cannot be detected by PTR-ToF-MS either, because the proton affinity of butane (153.7 kcal/mol (Estebes et al.; J. Phys. Chem. A 2000, 104, 26, 6233–6240)) is lower than that of water (165.3 kcal/mol (NIST)). This makes that the ionization of butane by reaction with $H_3O^+$, in which this technique is based, is not possible.

*It would be useful to include a figure of the photolysis lamp spectrum in the SI.*

**Authors' reply:** In the revised SI, the spectrum of the solar simulator will be included as Figure S2. This spectrum was measured by a spectroradiometer (Ocean Optics, model USB2000+) working with a CC-3-DA cosine corrector. The same figure will include the reference spectrum AM 1.5G of the Sun, corrected with the intensity measured in our experiments.

[Figure]

**Figure S2:** Irradiance spectrum of the ABA class solar simulator used in this work compared with the solar reference spectrum AM 1.5G, corrected with the intensity measured in this work.

*Product studies: The approach to product analysis described in the paper appears to assume RO2+RO2 radical chemistry. To the extent this is true, the results derived from these experiments may not be generally applicable to the atmosphere, where RO2 reactions with NO and with HO2 often dominate RO2 reactivity. However, it is also often true in systems where primary RO2 production greatly exceeds other primary radical production (eg, HO2 and NO), that HO2 chemistry remains important (due to substantial HO2 generation from initial RO2 + RO2 reactions).*

**Authors' reply:** There is no NOx in our chamber during our experiments, so the Reviewer is right: the chemistry of peroxyl ($RO_2$) radicals dominates the mechanism under the established conditions. We are aware that, in most locations, the chemistry involving NOx is of great importance, but in this paper, we wanted to look at the mechanism under NOx-free conditions, which we believe that is still of interest in non-polluted areas.

We agree with the reviewer that primary $RO_2$ concentration greatly exceeds $HO_2$ radicals in our system. $HO_2$ radicals are formed in reactions such as $RO + O_2$, and their concentration may be substantially lower than that for $RO_2$, similarly to what observed in other experimental setups (for example, Rissanen et al., J. Am. Chem. Soc. 2014, 136, 15596−15606). Although we did not mention it in the manuscript, we checked the role of the $RO_2+HO_2$ reactions in the formation of identified reaction products. General reaction channels for a $RO_2+HO_2$ reaction are:

$$RO_2 + HO_2 \rightarrow RO + OH + O_2$$
$$\rightarrow ROH + O_3$$
$$\rightarrow ROOH + O_2$$

For example, when $R=CH_3CH_2CH(CH_3)C(O)$, which is formed both in photolysis (channel R1a) and in the Cl reaction (channel R3b), the corresponding ROH, $CH_3CH_2CH(CH_3)C(O)OH$ was detected by PTR-ToF-MS and GC-MS, although found to be a negligible product, and the corresponding ROOH, $CH_3CH_2CH(CH_3)C(O)OOH$, was not detected with any of the techniques used in this work.

*HO2 + RO2 reactions often produce ROOH in high yield. What would be the fate of ROOH in your experimental system. Would ROOH decompose and be detected at other*

*products in this system/instrumentation.*

**Authors' reply:** According to several reviews (Jackson & Hewitt, Crit. Rev. Environ. Sci. Technol. 1999, 29:2, 175-228; Lee et al., Atmos. Environ. 2000, 34(21), 3475-3494) the possible pathways for destruction of ROOH in the atmosphere include photolysis, reaction with OH, or loss by physical deposition to the ground. In our chamber, the most likely fate of ROOH would probably be its reaction with Cl atoms. For example, the reaction channel for $CH_3CH_2CH(CH_3)C(O)OOH$, similarly to the OH-reaction, is expected to be:

$$CH_3CH_2CH(CH_3)C(O)OOH + Cl \rightarrow CH_3CH_2CH(CH_3)C(O)O_2 + HCl$$

Photolysis in our chamber is also likely, although according to Kleinman (J. Geophys. Res. 1986, 91, 10889–10904), it is a slower process than Cl-reaction:

$$CH_3CH_2CH(CH_3)C(O)OOH + h\nu \rightarrow CH_3CH_2CH(CH_3)C(O)O + OH$$

The $CH_3CH_2CH(CH_3)C(O)O_2$ and $CH_3CH_2CH(CH_3)C(O)O$ radicals that can be formed after the $CH_3CH_2CH(CH_3)C(O)OOH$ degradation are already present in our mechanism through the $RO_2 + RO_2$ route.

For any other ROOH in the mechanism, RO and $RO_2$ radicals are expected to be formed too, which yield the same products as explained in the mechanism proposed in the original manuscript by the $RO_2 + RO_2$ routes.

To avoid an overload of the depicted mechanism, we will change the "$+RO_2$" label in the mechanisms by "$+RO_2$ or $HO_2$" to include the channel in the $RO_2 + HO_2$ reaction in which RO radical is formed, and a short sentence will be added to the figure caption to include the possibility of ROOH formation and its fate.

*Can ROOH be specifically detected with the applied instrumentation?*

**Authors' reply:** PTR-ToF-MS can detect peroxides if their lifetime is high enough to reach the reaction chamber of the instrument. For example, Warneke et al. (J. Atmos. Chem. 2001, 38, 167–185) detected hydroxy-isoprene-hydroperoxides with a PTR-MS. In our study, the most probable ROOH's, according to the proposed reaction mechanism, i.e. $CH_3CH_2CH(CH_3)C(O)OOH$ and $CH_3CH_2CH(CH_3)OOH$, have been searched among the generated products by PTR-ToF-MS, which is the most sensitive analytical technique that we used. However, the expected ions $C_5H_{10}O_3H^+$ and $C_4H_{10}O_2H^+$, corresponding to $CH_3CH_2CH(CH_3)C(O)OOH$ and $CH_3CH_2CH(CH_3)OOH$, were not detected at m/z= 119 and 91, respectively.

*What is the RO2 lifetime in these experiments? What is the potential for autoxidation (RO2 H-shift) within the reaction chambers? Within the atmosphere? Suggest these questions should be discussed and clarified in the manuscript.*

**Authors' reply:** With our smog chambers, since we cannot detect radicals, we are not able to determine the $RO_2$ lifetime. As discussed in replies above, $RO_2$ degradation pathways in the atmosphere might be reactions with NO, $HO_2$, and $RO_2$. In real atmospheres, the $RO_2 + RO_2$ reactions are slower than the $RO_2 + HO_2$ and $RO_2 + NO$ reactions, but, in our system, the dominating radical is $RO_2$ since NO was not added to the mixture and $HO_2$ requires a more complicated chemistry than $RO_2$ to be formed. So, in our chamber, where $RO_2$ chemistry might be dominating, $RO_2$ lifetime will probably be longer than in real atmosphere.

The $RO_2$ autoxidation consists of an intramolecular H-abstraction or a H-shift, that generates and alkyl radical, R, that has a hydroperoxyl functional group (-OOH). This alkyl radical can even further react with $O_2$ to produce a new $RO_2$ radical. In the case of the $RO_2$ radicals formed in the Cl reaction of 2MB, the most likely H atom that can be shifted is the one of the aldehydic group, similarly to what was observed in one of the $RO_2$ radicals formed in the OH+methacrolein reaction (Crounse et al., J. Phys. Chem. A 2012, 116, 5756−5762). For example, in the Cl+2MB reaction, the $RO_2$ radical formed in channel R3b can be autoxidized to:

$$CH_3CH_2CH(CH_3)(OO)C(O)H \rightarrow CH_3CH_2CH(CH_3)(OOH)C(O)$$

The $CH_3CH_2CH(CH_3)(OOH)C(O)$ radical can decompose, as proposed by Crounse et al., to form butanone, CO and OH radicals:

$$CH_3CH_2CH(CH_3)(OOH)C(O) \rightarrow CH_3CH_2C(O)CH_3 + CO + OH$$

Then, detected butanone and CO can also be formed through this pathway. They were already included among the products observed through channel R3b, in the $RO_2 + RO_2$ reactions. Therefore, the inclusion of the $RO_2$ autoxidation in this channel in our mechanism will not change the conclusions of the manuscript. However, with the techniques used in this work, since we cannot detect $RO_2$ radicals, it is not possible to differentiate if the observed products are generated by the $RO_2 + RO_2$ reaction or by autoxidation.

*A simple kinetic simulation of the chemistry occurring within reaction chambers may be highly useful for a better understanding of the RO2 chemistry in these experiments.*

**Authors' reply:** The chemistry occurring within the reaction chamber, as depicted in the reaction mechanism presented in the manuscript, is very complex and involves a great number of reactions. A *simple* kinetic simulation would not be accurate enough to understand the $RO_2$ chemistry since, as far as we know, the rate coefficients for reactions involving most of the $RO_2$ formed in this work, like $CH_3CH_2CH(CH_3)C(O)O_2$, $CH_3CH_2CH(CH_3)O_2$,or $CH_3CH_2C(CH_3)(O_2)C(O)H$, are still unknown.

*SOA studies: Related to the comment above – to be useful for modeling the atmospheric fate of these species, SOA formation needs to be quantified for well-known conditions (including, RO2 fate [fraction reacting with RO2, HO2, and NO] and RO2 lifetime [can autoxidation be important, RO2 H-shift]) which relate to those found in the atmosphere. Differences in these pathways can substantially alter the formation of SOA, and as such, at a minimum one needs to carefully describe the experimental RO2 fate when reporting an SOA yield.*

**Authors' reply:** We agree with the Referee that it is important to know the specific conditions at which the SOA yield is quantified. In this case, we only know for sure what the initial concentrations of 2MB and $Cl_2$ in the chamber are. As explained above, none of our instruments can detect the presence of species like the $RO_2$ radicals. We understand that our results are of limited use in models that try to reproduce the behaviour of 2-methylbutanal in the atmosphere, but the aim of this work was to show that the SOA formation in its Cl reaction is possible, although in a very low extent.

*The quantification of particulate matter is a challenge, even, in large chambers, requiring careful correction for size dependent wall-loss, and correction for vapor-particle-wall partitioning of semi-volatile species. If these corrections have been done here this should be described in more detail.*

**Authors' reply:** Corrections were done to account for the loss of particles and gaseous 2MB onto the reactor walls. Since this was already described in one of our previous works (Antiñolo et al., Atmos. Environ. 219, 117041, 2019), we

decided to skip this part to focus on the results of this work.

The experimental aerosol mass ($M_t$) was corrected to obtain $M_{SOA}$ by means of the following equation, in which $t$ is the time after the Cl reaction started:

$$M_{SOA}=M_t(1+k_{SOA\ loss}\ t) \tag{1}$$

The concentration of reacted 2MB, $\Delta[2MB]$, at every reaction time $t$ was calculated using the following equation:

$$\Delta[2MB] = [2MB]_{0,corrected}-[2MB]_t \tag{2}$$

where $[2MB]_t$ is the concentration at a given time of 2MB and $[2MB]_{0,corrected}$ is defined as the initial concentration corrected with the wall loss:

$$[2MB]_{0,corrected}=[2MB]_0\exp(-k_{2MB\ loss}\ t) \tag{3}$$

where $[2MB]_0$ is the initial concentration introduced in the chamber.

*The fact that early nucleation occurs is interesting. It may be more fruitful to shift focus to this, describing potential mechanisms, and whether these may have atmospheric significance.*

**Authors' reply:** The SOA formation can be explained through a simple mechanism proposed by Kroll et al. (2007), and shown in Figure A, in which, after the oxidation of the parent VOC, a semi-volatile species is formed in the gas phase, $A_g$. This gas-phase species can undergo two different processes: partitioning to the particle phase, $A_p$ (and subsequent reaction in the particle phase to form $B_p$) or reaction to form other gas-phase species (X). If the partition to the particle phase is more important, there will be a very short induction period to generate SOA. Unfortunately, the gas-particle partition equilibrium constant, $K_p$, that can determinate in what extent partition to the particle phase is important, and that can be determined through the Odum et al. equation could not be determined in this work because, after *ca.* 12-30 minutes, the formed particles were too big ($D_p >0.52$ μm) to be detected by our FMPS spectrometer.

[Figure]

Figure A. Mechanism for SOA formation from a single semi-volatile species $A_g$ proposed by Kroll et al. (2007).

*It may be helpful to add a Table to the SI for yields of the more certain products from photolysis and Cl from each quantification method.*

**Authors' reply:** We will add the table below to the SI as suggested by the Reviewer, so the presentation of the results can be more easily understood. It will be included as Table S3.

**Table S3.** Summary of the product yields obtained in this work.

| Cl reaction | | |
|---|---|---|
| **Compound** | **Detection method** | **Yield / %** |
| Butanone | FTIR | $49.3 \pm 0.8$ |
| | PTR-ToF-MS | $32.4 \pm 0.2$ |
| Acetaldehyde | FTIR | - |
| | PTR-ToF-MS | $58.4 \pm 0.2$ |
| Methylglyoxal | FTIR | - |
| | PTR-ToF-MS | $13.5 \pm 0.2$ |

| Photolysis reaction | | |
|---|---|---|
| **Compound** | **Detection method** | **Yield / %** |
| Butane | FTIR | $9.80 \pm 0.31$ |
| Butanone | FTIR | $14.8 \pm 0.5$ |

*While not obtrusive to the point of obscuring understanding, the sentence structure could be improved in several places to improve clarity and readability (e.g. LN62-63, LN113-114)*

**Authors' reply:** Some sentences have been rephrased in the revised manuscript following the Reviewer's suggestion.

**Reply to specific comments**

*LN14: for Cl rate coefficient should be 'x10^(-10)'?*

**Authors' reply:** This typo will be corrected in the revised manuscript.

*LN31: I'm not familiar with O3 reactions with saturated aldehydes. Could you include more description and references? NO3 radical is known to react with aldehydes, thru aldehydic-H abstraction. Might be useful to mention this in overview.*

**Authors' reply:** Generally, only the reaction of $O_3$ with unsaturated compounds is fast enough to be important since the favoured mechanism is the $O_3$ addition to the double bond. In fact, $O_3$ reaction of saturated aldehydes were reported to be extremely slow ($<10^{-20}$ $cm^3$ molecule$^{-1}$ s$^{-1}$ at 298 K), and, therefore, of negligible importance in the atmosphere, by Atkinson and Carter (Chem. Rev. 1984, 84, 5, 437–470). We will include a mention to this in the revised manuscript.

On the other hand, $NO_3$ reaction of 2-methylbutanal is quite important in the atmosphere at night, with a rate coefficient of $(2.56 \pm 0.49)$ $10^{-14}$ $cm^3$ molecule$^{-1}$ s$^{-1}$ (Cabañas et al., Phys. Chem. Chem. Phys., 5, 112-116, 2003). However, the aim of this work was to discuss the **diurnal** degradation of 2-methylbutanal.

*LN67: It would be useful to state the composition of diluent gas (and other places in manuscript)*

**Authors' reply:** The diluent gas used in this mixture was synthetic air (AirLiquide, 99.999 %), that contains 20% $O_2$, $\leq$ 1 ppm CO, $\leq$ 1 ppm $CO_2$, $\leq$ 3 ppm $H_2O$, $\leq 0.1$ ppm $C_nH_m$, and the rest is $N_2$.

We will add to the manuscript the subsection 2.4. Chemicals, in which we will give details, not only of the diluent gas, but of other chemicals used in this work.

*LN85-98: Often, as a result of chemistry following radical formation from initial photon excitation, additional oxidants can be formed (eg, OH/HO2) either directly or from*

*photolysis of resulting products. These oxidants can also contribute to the loss of the starting material. Were any efforts made to constrain the impact such chemistry may have on these results?*

**Authors' reply:** An experiment of photolysis of 2MB was done in the presence of cyclohexane ([cyclohexane]/[2MB] = 8.2) which is widely used as OH radical scavenger in this kind of experiments. The photolysis quantum yield obtained in the presence of cyclohexane ($\Phi_{cyclohexane}$ = 0.29 ± 0.02) showed no difference when compared with the reported in the manuscript ($\Phi_{eff}$ = 0.30 ± 0.05), indicating that the chemistry of other oxidants is negligible. This will be added in the revised manuscript.

*LN94: insert 'large' before 'surface'?*

**Authors' reply:** We will make this change following the Referee's advice.

*LN115-139: It would be useful to state the volume (or gas residence time) of the LIF setup*

**Authors' reply:** The gas residence time in the LIF reactor varies depending on the total gas flow, pressure, and temperature. Considering the geometry of the reactor, in our experiments, the residence time ranges from 0.8 to 1.1 seconds (at 353 and 263 K, respectively) at 50 Torr of total pressure, from 4.9 to 6.6 seconds (at 353 and 263 K, respectively) at 300 Torr of total pressure, and from 9.8 to 13.2 seconds (at 353 and 263 K, respectively) at 600 Torr of total pressure. In the revised manuscript, the following sentence will be added: "The residence time in the cell ranged between 0.8 and 13 s, depending on the flow, pressure and temperature conditions of the experiment".

*LN177-188: It would be good to explicitly specify the diluent gas, reaction pressure, and temperature.*

**Authors' reply:** The Referee is right, the diluent gas (synthetic air) and the P,T conditions of the experiments (760 Torr and 298 K) were not stated in Section 2.3.2, as they are the same as in the kinetics. All this information was given in the previous section. However, to make it clearer, we will include it again in the

product study for the Cl reaction.

*LN239-241: It may be clearer to write this section for wavelength region used for the photolysis expts (>290nm?). Are all three channels listed here accessible under these photon energies? DeltaG estimate?*

**Authors' reply:** We agree with the Reviewer that it is clearer if we state that photolysis occurs at $\lambda \geq 290$nm instead at $\lambda = 220-360$ nm, as previously written in the original manuscript. This change will be made in the revised version.

$\Delta G^0_{298 \text{ K}}$ were calculated, as explained above, and the three channels are accessible at $\lambda \geq 290$nm.

*LN249-250: Can you discuss this further? Why is the CO yield dependent on initial MB?This is may be telling you something worth digging into.*

**Authors' reply:** There are several studies that observed changes in product yields when changing experimental conditions like relative humidity, temperature or concentration of reactants. For example, in the Friedman and Farmer paper (Atmos. Environ. 2018,187, 335-345), they observed that the yields of the organic acids formed during the OH reaction of monoterpenes changed as a function of [OH]. Their explanation was that, when their concentration is higher, OH radicals oxidized further the acids, causing a decrease in their yield. In our case, this explanation is not valid since we observed that the CO yield is increased when the initial concentration of 2MB is lower. Looking more carefully to the graphs from which $Y_{CO}$ are determined in this work (see Figure B below), it can be observed that the maximum [CO] generated in the experiments is very similar, and the only thing that changes is the amount of 2MB reacted. This indicates that CO is coming from a process that takes place independently of the initial 2MB, and the most probable process occurring in our photolysis cell is CO generation from the walls. This probable origin of CO will be included in the revised manuscript.

[Figure]

Figure B. Plots from which $Y_{CO}$ were determined in this work.

*LN270-275: Can butane also be measured by the PTRMS or GC-MS? If so, it would be worthwhile to describe these results, as they would lend confidence to the FTIR quantification of butane.*

**Authors' reply:** Unfortunately, butane could not be detected neither by GC-MS or PTR-ToF-MS, as described above. Therefore, its quantification was only possible by FTIR.

*Fig7: Is that CO2 initial and 50 min spectra? Is this understood? Seems like there is also H2O in the residual spectra? Is this understood? Can impact of OH chemistry in the Cl expts be quantified using this?*

**Authors' reply:** The presence of unexpected $CO_2$ and $H_2O$ in our initial FTIR spectra is due to the fact that the FTIR spectrometer is not purged with $N_2$ and there are changes in the $CO_2$ and $H_2O$ concentration in the lab that can affect the recorded spectra. In addition, the background spectrum, that can remove the $CO_2$ and $H_2O$ features, was recorded at the beginning of a series of measurements that includes dark reactions and photolysis tests. For the initial spectrum in Fig. 7, the

background was, at least, 90 min old.

*LN350-363 &Fig8: More discussion of how PTR signals for acetald and 2MB were deconvoluted is needed.*

> **Authors' reply:** 2MB is ionized in our PTR-ToF-MS into two different ions: $C_5H_{10}OH^+$ and $C_2H_4OH^+$. The $C_2H_4OH^+$ ion is formed also by acetaldehyde, so the signal due to 2MB and the signal due to acetaldehyde had to be separated to correctly quantify 2MB from the $C_2H_4OH^+$ ion. The way we managed to do this was very simple. First, only 2MB was present in the sample, so the $C_2H_4OH^+$ ion was only coming from 2MB, and it was possible to establish the amount of the $C_2H_4OH^+$ and $C_5H_{10}OH^+$ ions (61% and 39%, respectively), and the ratio between them (1.56). When the reaction started, it was possible to calculate the concentration of the $C_2H_4OH^+$ ion due to 2MB ($[C_2H_4OH^+]_{2MB}$) considering the $C_5H_{10}OH^+$ ion and its relationship with the $C_2H_4OH^+$ ion:
>
> $$[C_2H_4OH^+]_{2MB} = 1.56\,[C_5H_{10}OH^+]$$
>
> Once $[C_2H_4OH^+]_{2MB}$ was known, it was subtracted from the total concentration of the $C_2H_4OH^+$ ion ($[C_2H_4OH^+]_{Total}$) to determine the concentration of acetaldehyde ($[CH_3C(O)H]$):
>
> $$[CH_3C(O)H] = [C_2H_4OH^+]_{Total} - [C_2H_4OH^+]_{2MB}$$

*Also, there should be some explanation for how PTR calibrations were conducted.*

> **Authors' reply:** The products were quantified by using the transmission calibration of the instrument, so the software converts the signal into concentration. This transmission calibration was performed by using a standard gas mixture: TO-14A Aromatics Mix (14 components) (Scott/Air Liquide).

*LN396: 'J' --> 'H' ?*

> **Authors' reply:** This typo will be corrected in the revised manuscript.

*LN431: Probably more realistic to assume [Cl] scales with sun (J(MB)?) and then scale to constraint (1.3e5)? This will impact discussion LN448-449.*

> **Authors' reply:** We agree with the Reviewer that $[Cl]_{peak}$ concentration is not

realistic for the whole day, and that is why we stated in the manuscript that its contribution should be considered as an upper limit. Moreover, we considered the same concentration at the end of the day, when the [Cl] concentration is known to be much lower. For the first hours of the day, $[Cl]_{peak}$ could probably work fine, but the impact is surprisingly (and erroneously) high after 18:00. As far as we know, there is no hourly measurements of [Cl] in Valencia (Spain) or a similar location. That makes hard to provide an accurate hourly contribution of Cl atoms to the loss of 2MB. From 20:00 h, obviously solar radiation, that can generate both Cl atoms and OH radicals, is less intense. Therefore, we will remove the relative contributions of OH, Cl, photolysis at 20:00 and 22:00 in Figure 9. Anyways, we will keep them for the rest of the day, taking into account that, even using $[Cl]_{peak}$, the major tropospheric route for the homogeneous loss of 2MB is the OH-reaction. Figure 9 will be replaced by this one:

[Figure]

**Figure 9**. Relative contribution of the three removal routes studied in this work to the total diurnal loss.

*LN447-452: It would be useful to report the daily integrated loss pathways considered here for this (ie over day fraction 2MB lost to OH, Cl, photolysis).*

**Authors' reply:** Following the Referee's suggestion, we will include in the revised manuscript the estimation of the tropospheric lifetime considering the average concentrations in 24 h of OH and Cl, and averaged values of the solar

spectral actinic flux for 24 h in Valencia. Considering 24 h-average values of [OH] ($1 \times 10^6$ radicals cm$^{-3}$, Krol et al., (1998)), [Cl] ($1 \times 10^3$ atoms cm$^{-3}$, Singh et at. (1996)), and $F(\lambda,z,\theta)$ for Valencia in June, $\tau$ can be estimated as 7.5 h for a 24 h-period, with a relative contribution of 73% of the OH-reaction, 27% of photolysis, and less than 1% of the Cl-reaction.

*Fig S1: State if units are log10 or loge A.*

> **Authors' reply:** Although the UV absorbance measured by the instrument is defined in base 10, the UV absorption cross sections are usually expressed in base e. For that reason, the absorbance in the UV region is expressed in base *e* throughout the manuscript. This will be clarified in the revised manuscript.

*Table S1: State more precisely the wavelength intervals used. Eg. 1-nm integrated intervals with midpoint wavelength listed in table?*

> **Authors' reply:** The $\sigma_\lambda$ values presented in Table S1 are those determined at the stated wavelength. They are not integrated $\sigma_\lambda$, but absolute values. The spectral resolution of the spectrograph was higher than 0.11 nm, but in Table S1 $\sigma_\lambda$ were listed every 1 nm for ease of presentation. We will include an excel file with all determined $\sigma_\lambda$ between 220 and 360 nm in the SI. This has been clarified in the text of the main manuscript and in the Table S1 heading.

*Fig S6: There seems to be some issues with x-axis alignment on this figure. Suggest these should either be perfectly aligned (ie automatically, using plotting program) or the numbers for each x-axis should be retained. For example, butane and acetaldehyde ticksto not line up with others (butane C-H band should be aligned with that of butanol). Are there CO2 bands in some of these spectra (eg butanone, acetald., butanol, and MGLX)? Also H2O in methylglyoxal? If so, probably worthwhile to point out these as impurities, not considered in fitting.*

> **Authors' reply:** The misalignment in x-axis of Fig S6 has been solved.
> Reviewer is right that there are $CO_2$ and $H_2O$ impurities in some of the spectra. The presence of these atmospheric gases was discussed in a aforementioned reply. For methylglyoxal, note that this spectrum was taken from the EUROCHAMP database (Ródenas, 2017), not recorded in our lab. Thus, we do not know the

reason for the presence of $H_2O$ and $CO_2$ in that spectrum. This will be mentioned in the caption of this figure in the revised version of the SI. This figure has been changed to:

[Figure]

**Figure S6:** FTIR spectra reference used in this work. Butanone, formaldehyde, acetaldehyde, propanal, and butanol spectra were recorded in our lab. Butane spectrum was taken from the NIST FTIR database (Linstrom and Mallard, 2018), and methylglyoxal spectrum was taken from the EUROCHAMP database (Ródenas, 2017). $CO_2$ and $H_2O$ are present as impurities in some of these spectra and were not considered.

---

## Referee Report (RR1)

**Manuscript ID:** acp-2021-758 (2$^{nd}$ Revision)

**Authors:** María Asensio, María Antiñolo, Sergio Blázquez, José Albaladejo, Elena Jiménez

**Ms. Title:** Evaluation of the daytime tropospheric loss of 2-methylbutanal.

**General Comments:**

The authors did an excellent work to address all the suggestions/corrections and comments and the present reviewer believes that the present form of the manuscript meets the high standards of ACP and can be published as is. The only remaining issue is related to tables and graphs units presentation. A minor comment that also noted previously, it would be easier for the reader to see the units in parenthesis without math signs, e. g., instead of Wavenumber /cm$^{-1}$, use Wavenumber (cm$^{-1}$). However, this is a minor technical change and does not affect the high quality of the paper.

**Suggestion**: Published as is

---

## Author Response (AR2)

**REVIEW BY EDITOR**

We kindly thank the editor for considering this manuscript for publication and for the following comments that will help us to improve the quality of the paper.

1) Line 365, and Figure S9/Table S4: It doesn't appear to me that any correction has been made to the product concentrations for their loss due to reaction with Cl-atoms. Please apply these corrections to give the 'corrected' yield data.

**Authors' reply:** The editor is right: the reported yields in the original manuscript were not corrected to account for the product loss due to Cl reaction. The concentration of the products can be corrected by means of a factor given by equation (1) as in previous works (e.g., Ceacero-Vega et al. J. Phys. Chem. A, 116, 4097–4107, 2012):

$$F = \frac{k_{2MB} - k_{prod}}{k_{2MB}} \times \frac{I - \frac{[2MB]_t}{[2MB]_0}}{\left(\frac{[2MB]_t}{[2MB]_0}\right)^{k_{prod}/k_{2MB}} - \frac{[2MB]_t}{[2MB]_0}}$$
(1)

where  $k_{2MB}$  is the rate coefficient for the 2MB + Cl reaction determined in this work and  $k_{prod}$  is the rate coefficient of the Cl-reaction of the product:  $k_{butanone} = 4 \times 10^{-11}$ cm3 molecule-1 s-1 (IUPAC recommendation),  $k_{acetaldehyde} = 8 \times 10^{-11}$  cm3 molecule-1 s-1 (IUPAC recommendation), and  $k_{methylglyoxal} = 4.8 \times 10^{-11}$  cm3 molecule-1 s-1 (Green et al., Int. J. Chem. Kinet., 22, 689-699, 1990). The corrected product yields can be determined from the slope of the plot of the product concentration corrected with the *F* factor *versus*  $\Delta$ [2MB]. Figures A and B show the non-corrected and corrected product yield plots for the FTIR and PTR-ToF-MS techniques, and Table A summarizes the results obtained without and with correction. In the revised manuscript, the reported product yields in Table S4 and Figure S9 are those corrected with *F* factor.

Figure A. Comparison between the non-corrected and corrected product yield for butanone quantified by FTIR.

Figure B. Comparison of the non-corrected and corrected product yields for acetaldehyde, butanone, and methylglyoxal quantified by PTR-ToF-MS.

| Compound      | Detection method | Non-corrected
Yield (%) | Corrected
Yield (%) |
|---------------|------------------|----------------------------|------------------------|
| Butanone      | FTIR             | $49.3\pm0.8$               | $53.1\pm1.6$           |
|               | PTR-ToF-MS       | $32.4\pm0.2$               | $34.9\pm0.6$           |
| Acetaldehyde  | FTIR             | -                          | -                      |
|               | PTR-ToF-MS       | $58.4\pm0.2$               | $67.9\pm 0.8$          |
| Methylglyoxal | FTIR             | -                          | -                      |
|               | PTR-ToF-MS       | $13.5\pm0.2$               | $14.8\pm0.2$           |

Table A. Summary of the product yields.

As can be seen both in Figs A and B, and in Table A, the differences between the corrected and non-corrected yields are quite small, between 1 and 10%, and that is the reason why the correction was not applied in the submitted manuscript. Since the editor thinks that the correction should be included, we have modified Fig. S9 and Table S4 to account for this correction, and a sentence is included in the manuscript to indicate that a correction was done.

2) Near Line 411: I agree that the reaction at the 1- and 2-position likely are most important but I am not sure that there is sufficient evidence here to confirm that, as methylglyoxal may not be a major product of reaction at the CH2 group? Perhaps some re-phrasing of the sentence beginning on Line 411 would be helpful?

**Authors' reply:** The sentence the editor referred to has been rephrased as follows in the revised manuscript:

The product yields obtained in this work indicate that the channels for the H-abstraction from the - C(O)H group and the tertiary H in C-2 are more favoured, given that  $Y_{butanone}$  is more than 2 times higher than  $Y_{methylglyoxal}$ . In addition, methylglyoxal may not be a major reaction product from H-abstraction from the C-3 site.

*3)* In the mechanism, the CH3CH2O radical will react with O2 to make CH3CHO, not decompose - please change this throughout.

**Authors' reply:** The decomposition of the CH3CH2O radical has been changed throughout the mechanisms (Figs. S13 and S14) by its O2 reaction to form CH3C(O)H and HO2.

Lastly, I note a few places where grammatical improvements can be made. Please replace the current text with the words/phrases listed below.

Authors' reply: The small changes concerning the grammatical improvements suggested by the editor have been considered in the new revision of the manuscript.

**REPLY TO REFEREE #1**

We thank the Referee for the comments and suggestions made to improve the manuscript.

The only remaining issue is related to tables and graphs units presentation. A minor comment that also noted previously, it would be easier for the reader to see the units in parenthesis without math signs, e. g., instead of Wavenumber /cm-1, use Wavenumber (cm-1). However, this is a minor technical change and does not affect the high quality of the paper.

Authors' reply: As the reviewer said, the suggested change is indeed a minor technical issue. Even though the IUPAC recommends the use of "/" to separate the magnitude and its units in tables and the legend of graphs, we have replaced "/" by "( )" for ease of reading in all tables and figures of the revised manuscript and supporting information.

**REPLY TO REFEREE #2**

We really appreciate the Referee's comments and have tried to address them successfully.

Cross-sections for product quantification: To extend the impact the these observations, it would be useful to include the absolute IR cross-sections used to quantify the 2-MB and the oxidation products in this work. This could be done in a number of ways. For example, the Y-axes of Figure S6 could be shown in cross-section units, or a Table of integrated band cross-sections could be included, or single point cross-sections could be given for strong lines (with instrumental optical parameters).

Authors' reply: In the present work, the absolute IR absorption cross sections at each wavenumber were determined from the slope of the Beer-Lambert's plot. As this information can be useful for other researchers to quantify 2MB, we have included an Excel file with all  $\sigma_v$  in the supporting information. If needed by other researchers, they can easily use the tabulated IR absorption cross sections to obtain the integrated cross sections in the desired wavenumber range, or even to extract the single point IR absorption cross sections for strong lines. For reaction products like HCl, CO, etc, we do not consider appropriate to include their IR integrated band intensities since they have been published elsewhere or were taken, as for methylglyoxal, from open databases.

Butane: Still surprised that the small yield of butane can be determined so well (to  $\sim 3\%$  uncertainty) using the FTIR in the face of the overlapping bands from other species. Could you provide more insight into this. Perhaps add butane to Figure S9, and show an example spectral fit?

**Authors' reply:** As shown in the IR spectra below, although the absorption bands of butane overlap with butanone, it was possible to quantify it. In the upper panel, the residual spectrum obtained after irradiation and subtraction of unreacted 2MB is shown. The IR bands around 3000 cm-1 and 1500 cm-1 come from butane (reference spectrum in central panel). After subtracting the IR features from butane, the bands due to CO and butanone are still visible.

---

## Author Response (AR3)

We would like to thank the editor for the further revision that will certainly improve the quality of the manuscript.

*For line 413, the point is that, since methylglyoxal may not be a major product from the C-3 site, you cannot conclude from the observations that aldehydic and tertiary C-2 abstraction are dominant. I suggest wording such as the following be employed - "Although abstraction at the –C(O)H and tertiary C-2 site are likely dominant, methylglyoxal may not be a major product of C-3 and thus our product data are not conclusive on this issue."*

**Authors' reply:** As suggested by the editor, we have reworded the text in lines 410-413.

*Figure S6: The referee is asking that absolute cross sections be given for the species measured as part of the work (e.g., butanone, acetaldehyde, formaldehyde, …), not just for the parent 2MB. As suggested, perhaps absolute cross sections instead of absorbance units for the relevant species in Figure S6 would be the easiest approach to this?*

**Authors' reply:** As suggested by the Reviewer and the editor, we have changed Figure S6 to show the absolute absorption cross sections of butanone, formaldehyde, acetaldehyde, butane, 2-butanol, methylglyoxal, propanal, HCl, and CO, instead of absorbances.

[Figure]

New Figure S6

*Butane quantification: It seems to me that the better approach to Figure C would be to follow the procedure that is outlined in the text – i.e., subtract the 2MB features, then the butanone, and then compare the residual to the butane spectrum (perhaps by showing the residual with and without butane removed).*

**Authors' reply:** We agree with the editor that the procedure followed for the subtraction of butane will be easier to understand if the figure shows the same order.

[Figure]

Figure. Set of spectra showing the subtraction procedure to obtain the butane concentration as described in the text.

Therefore, we present here the following spectra:

    A.   Final spectrum after 150 min of photolysis.

    B.   A spectrum - 2MB.

    C.   B spectrum - butanone.

D. Butane spectrum.

E. C spectrum - butane.

---

## Author Response (AR4)

We greatly appreciate the following comment on the IR absorption cross sections presented in Figure S6.

**Editor's comment**: *I have looked at Figure S6 and have some misgivings with the cross sections - values for HCHO and CO both seem very low to me (compared at least to what is used in our laboratory). Can you please check all of the cross section values in the figure, make sure they are being reported correctly, and indicate whether they are base 10 or base e. I think this is critical piece to assessing the reported product yields.*

**Author's reply**: The IR absorption cross sections reported in Figure S6 were expressed in base 10. To unify the way of expressing both the UV and IR absorption cross sections, all reported σ in the revised manuscript are now in base e. This has been indicated in the revised text.

Regarding the low **absorption cross sections of CO**, the editor is totally right. When we initially merged the IR spectra of CO and HCl, at some point before submitting the paper, the scaling factor was deleted. We are thankful for pointing this error out that we have corrected. Now, in Figure S6 this scaling factor (divided by 5) was added in the figure. We also want to highlight that CO IR spectrum is highly structured and the spectral resolution may lead to differences in the measured absorption cross sections. The reference IR spectrum used in our work for identification, not quantification, of CO was recorded at 1 cm$^{-1}$ spectral resolution.

Concerning the absolute values of the **IR absorption cross sections of formaldehyde**, after changing the base 10 to base e, we have compared them with those calculated from two single IR spectra recorded in EUROCHAMP project (Ródenas et al., 2017) with the same resolution as that used in this work (1 cm$^{-1}$). Our values are around 2 times lower than the EUROCHAMP values. Note that the latter spectra present large absorption from $CO_2$ and $H_2O$ below 2400 cm$^{-1}$. So, we have only compared the 3100-2600 cm$^{-1}$ band.

[Figure]

[Figure]

  The source of this discrepancy can be related with the accuracy in knowing the formaldehyde concentration, which was synthesized from paraformaldehyde. For that reason, we were reticent to provide the absolute absorption cross sections in Figure S6. *Nevertheless, as we just use this spectrum for identification, the absolute values would need to be refined when needed for quantification*. To report more accurate IR absorption cross sections, as said before, it is better to follow the same procedure as that used for butanone and 2MB (see below).

**IR absorption cross sections for quantification of 2MB and butanone**

2MB and butanone are the two only compounds for which their IR absorption cross sections had to be determined in our lab to be quantified in this work. Their IR absorption cross sections (**in base e**) reported in the manuscript are plotted in the following figures.

[Figure]

These values were determined from the slope of Beer-Lambert plots considering NINE IR spectra for 2MB and SIX for butanone, as shown in the plots below. This procedure provides more accurate IR absorption cross sections than those obtained from a single IR spectrum.